



# Modelling sun-induced chlorophyll fluorescence (SIF) in evergreen conifer forests with a terrestrial biosphere model

Tea Thum[1], Javier Pacheco-Labrador[2], Mika Aurela[1], Alan Barr[3], Marika Honkanen[1], Bruce Johnson[3], Hannakaisa Lindqvist[1], Troy Magney[4], Mirco Migliavacca[5], Zoe Amie Pierrat[6], Tristan Quaife[7], Jochen Stutz[8], and Sönke Zaehle[9]

[1]Finnish Meteorological Institute, Helsinki, Finland
[2]Environmental Remote Sensing and Spectroscopy Laboratory (SpecLab), Spanish National Research Council (CSIC), Madrid, Spain
[3]University of Saskatchewan, Canada
[4]Department of Forest Management, University of Montana, Missoula, MT, USA
[5]European Commission, Joint Research Centre, Ispra (VA), Italy
[6]Jet Propulsion Laboratory, Pasadena, California, USA
[7]National Centre for Earth Observation, University of Reading, Reading, The United Kingdom
[8]University of California Los Angeles, California, USA
[9]Max Planck Institute for Biogeochemistry, Jena, Germany

**Correspondence:** Tea Thum (tea.thum@fmi.fi)

**Abstract.** Solar-induced chlorophyll fluorescence (SIF) is a small light signal emitted during the initial steps of photosynthesis and can be observed across scales (from photosystem level to satellites). To be able to model SIF, we need to understand the mechanistic processes (including both physical and biological) leading to the observed SIF signal. In this work, we implemented a representation of SIF emission and transmission processes into the terrestrial biosphere model QUINCY ('QUantifying Interactions between terrestrial Nutrient CYcles and the climate system'). We tested the model across three different boreal coniferous forests located in North America and Europe that have eddy covariance derived $CO_2$ fluxes and tower-based SIF observations. We find that alternative SIF radiative transfer approaches (one based on mSCOPE, one on two-stream radiative transfer model L2SM, and one empirically based) overestimate the SIF signal, but show no large differences in the timing of their seasonal and diurnal predictions. The two-stream radiative transfer model approach, L2SM, provided stable performance while being comparatively computationally efficient. We find that our parameterization for sustained non-photochemical quenching is important for successfully simulating the timing of the SIF seasonal cycle. However, our parameterization did not work equally well across all three sites, likely because of different temperature regimes at the sites. We further evaluated the potential of remote sensing -based SIF from TROPOMI (the TROPOspheric Monitoring Instrument) to provide accurate information on SIF and found that it can potentially be used in model development. This study illustrates the usefulness of observations at different spatial scales and the linkages between SIF and GPP and their seasonal development at three different evergreen forest sites.



## 1 Introduction

The northern latitudes are experiencing stronger climatic change than the rest of the globe (Rantanen et al., 2022). Boreal
forests located in these regions are an important part of the global carbon cycle and the biomass located in boreal forests is
estimated to be $264.9 \pm 10.5\,\mathrm{PgC}$ (Pan et al., 2024). Boreal forests are characterized by strong seasonality in environmental
conditions, with harsh winter conditions and shoulder seasons when air and soil temperature and light availability drive the
spring recovery and autumn drawdown of vegetation (Tanja et al., 2003; Thum et al., 2009; Vesala et al., 2010). The photosyn-
thetic activity of evergreen forests in these ecosystems cannot be easily tracked by reflectance-based remote sensing alone, as
the greenness is partially decoupled from the rate of photosynthesis (Walther et al., 2016). Solar-induced chlorophyll fluore-
scence (SIF) observations have proven to be more reliable proxies for tracking photosynthesis in these ecosystems (Pierrat et al.,
2024). Challenging conditions have led evergreen trees to develop different coping mechanisms. Sustained non-photochemical
quenching (NPQ) is one of them and it increases in winter, at the same time as the capacity of photosystem II decreases (Adams
et al., 2014). NPQ is a pH-independent mechanism associated with the retention of the xanthophyll cycle pigments zeaxanthin
and antheraxanthin and allows the needles to dissipate the incoming radiation as heat (Demmig-Adams et al., 2014).

Sustained NPQ can only be estimated from the active chlorophyll fluorescence (ChlF) observations, i.e. when a set of
saturating light pulses are delivered to a leaf under dark- and light-adapted conditions. Therefore, it cannot be directly obtained
from passive SIF observations that take place under natural illumination conditions, although progress is being made towards
optical sensing of NPQ (Van Wittenberghe et al., 2024). Including description of sustained NPQ in large scale terrestrial
biosphere models (TBMs) was started by Raczka et al. (2019), who used the state of acclimation, represented by a delayed
temperature sum developed by Mäkelä et al. (2004) in the parameterization. Climate-induced changes will alter the seasonal
cycle of vegetation, and the ability to have optical data to track photosynthetic activity is very helpful in understanding the
changes in the carbon cycle. Another feature of boreal forests is that they contain a lot of their carbon belowground (Bradshaw
and Warkentin, 2015). The use of terrestrial biosphere models (TBMs) makes it possible to study the entire carbon balance and
is therefore an important tool for studies at high latitudes.

Space-based observations have the ability to monitor the entire Earth's surface, and advances in remote sensing methods
and satellite technology are providing more data streams that can be used in carbon cycle studies (Schimel et al., 2019).
The ability to observe SIF from space has led to numerous applications (Mohammed et al., 2019). SIF is linked to the light
reactions of photosynthesis and can therefore provide information on terrestrial $CO_2$ uptake (Porcar-Castell et al., 2021). Early
research on SIF showed that the relationship between SIF and photosynthesis (gross primary productivity, GPP) is linear when
measured from space (Frankenberg et al., 2011; Sun et al., 2017). Subsequent work has challenged this assumption, showing
the relationship between SIF and GPP is more complex (Damm et al., 2015; Magney et al., 2020; Martini et al., 2022; Sun
et al., 2023b) even when using space-based observations (Balde et al., 2023). Therefore, process-based approaches are useful
for understanding the mechanistic drivers of the SIF-GPP relationship.

Observations of leaf-level chlorophyll fluorescence have been widely used in plant physiological research for decades.
Consequently, there is a thorough understanding of the mechanisms governing leaf-level fluorescence (Baker, 2008; Maxwell



and Johnson, 2000). When photons are absorbed by plant leaves, they have three main non-damage pathways: they can be used for photochemistry, emitted as ChlF, or dissipated as heat. Since these three pathways coexist, the amount of NPQ affects the relationship between ChlF and photochemistry. In ChlF, a small fraction of photons are re-emitted after giving up some of their

energy at higher wavelengths (SIF spectrum is between 650 and 840 nm, as in Fig. 1) (Porcar-Castell et al., 2021). SIF is ChlF that takes place under natural illumination conditions, and measuring it is referred to as a passive measurement of ChlF.

When moving from the leaf level to the canopy level, the interpretation of the measured signal becomes more challenging. Scattering and re-absorption of ChlF take place within the canopy (Van Der Tol et al., 2019). These processes influence how much of the SIF signal located in the red part of the spectrum is absorbed compared to the near-infrared (NIR) (also called "far-

red") region. The structural effects of the canopy play an important role in the transmission of the emitted SIF signal within the canopy (Paul-Limoges et al., 2018). The soil will also contribute to the SIF signal observed at the top of the canopy, as observed signals include contributions from both vegetation and soil components (Yang et al., 2025). The variability in radiative transfer through the canopy creates challenges for interpreting the measured SIF signal. By using radiative transfer and biological modelling, mechanistic drivers of the SIF signal can be disentangled, improving our interpretation of SIF (Damm et al., 2015).

The use of SIF in carbon cycle modeling has become widespread. The first leaf-level description for ChlF was in FluorMod (Miller et al., 2005). A wide-spread leaf level model that was further developed from FluorMod was within the Soil Canopy Observation of Photosynthesis and Energy fluxes (SCOPE) model (van der Tol et al., 2009). SCOPE is a site level model which combines the Farquhar photosynthesis model with a detailed radiative transfer scheme based on SAIL (van der Tol et al., 2009). A newer leaf level model, that was also implemented in SCOPE, was published a few years later (van der Tol

et al., 2014) and further developments have also been made (Vilfan et al., 2016, 2018). The SCOPE model has been widely used in many applications (e.g., Damm et al., 2015; Martini et al., 2019; Wang et al., 2023; Zhang et al., 2018). Recent model developments also allow the use of SIF to estimate GPP (Gu et al., 2019). These methods utilize the link between measured SIF and light reactions of photosynthesis and how these observations provide a link for actual electron transport from photosystem II to photosystem I. The model by Johnson and Berry (2021) has a tight coupling between photosynthesis and ChlF and allows

for bi-directional modelling, estimating SIF from GPP and vice versa.

TBMs are large-scale models used to study the biogeochemical cycles and land–atmosphere interactions. They can be run at a large scale (regional and global), but site-scale simulations are still possible. The modelling community has implemented SIF models in TBMs with varying degrees of complexity. For example, Koffi et al. (2015) augmented the Biosphere Energy Transfer Hydrology (BETHY) model with the full SCOPE model. Lee et al. (2015) used a simple scheme to account for

radiative transfer in the CLM implementation. Bacour et al. (2019) built a SCOPE emulator for an implementation in the ORCHIDEE model. These different approaches balance simplifying the complex physical phenomenon of radiative transfer in plant canopies against the length of the simulation time. However, full 1D radiative transfer based on the SCOPE model is too computationally demanding for many large scale applications (Sun et al., 2023a) and some modelling teams have needed to use parameterizations instead of the full model (Miyauchi et al., 2025). The computational burden becomes even more relevant

in different data assimilation approaches (Norton et al., 2019; MacBean et al., 2018). An empirical approach used in some studies would be worth investigating (Liu et al., 2020; Zeng et al., 2019) . A two-stream radiative transfer of SIF signal has



**Table 1.** The site characteristics of the three forests. LAI is one-sided.

| Abbreviation | Location (lat, lon) | Species and age (yrs) | LAI ($m^2\,m^{-2}$) | Air temp. (°C) | SIF instrument |
|---|---|---|---|---|---|
| CA-Obs | 53.99, -105.12 | Black spruce (>100) | 3.8 | 1.3 | PhotoSpec |
| FI-Sod | 67.36, 26.64 | Scots pine (90) | 1.3-1.4 | 0.3 | FloX |
| US-NR1 | 40.03, -105.55 | Mixed evergreen coniferous (>100) | 3.8–4.2 | 2.7 | PhotoSpec |

been considered to be computationally tangible (Sun et al., 2023a) and a recent model (Quaife, 2025) enables calculation of SIF, since it describes radiative transfer of emission originating from the canopy.

This work aims to improve the modelling of ChlF so that a TBM can fully exploit the potential of the different data streams associated with SIF and pave the way for data assimilation approaches. Our objectives were 1) to test different radiative transfer approaches for SIF and 2) to assess the role of sustained NPQ in modelling. The research questions of our study are therefore:

- Which radiative transfer model calculation methods were sufficiently robust for reliable SIF model predictions?

- How could we account for the influence of sustained non-photochemical quenching in the modeled SIF signal?

- What was the benefit of in-situ observations versus satellite observations of SIF in model development?

To answer these questions, we run simulations of TBM QUINCY ('QUantifying Interactions between terrestrial Nutrient CYcles and the climate system') and compared them with tower observations of SIF at three coniferous evergreen sites that experience a strong seasonal cycle with harsh winters. In addition, we tested how TROPOMI satellite data capture the seasonal cycle at one of these sites and how its magnitude differs from the site-level observations.

## 2  Materials and methods

### 2.1  Site descriptions and observations

The three study sites were Niwot Ridge (US-NR1) (Bowling et al., 2018; Burns et al., 2015; Magney et al., 2019a), USA, Saskatchewan (CA-Obs), Canada (Pierrat et al., 2021, 2022a) and Sodankylä (FI-Sod), Finland (Thum et al., 2007; Knorr et al., 2025). All of these sites are evergreen coniferous forests. The Canadian and Finnish sites are in the boreal zone, and Niwot Ridge is a subalpine forest. Further details about the sites are given in Table 1. All sites have eddy covariance flux observations as well as a tower-mounted in-situ SIF instrument. The Sodankylä site is part of the ICOS network (https://www.icos-cp.eu/) and the North American sites are part of AmeriFlux (https://ameriflux.lbl.gov/).

At the North American sites, SIF was observed with PhotoSpec (Grossmann et al., 2018) in two different spectral regions. The red region is between 680 and 686 nm, and the far-red region is between 745 and 758 nm. These observations were made with a 2D scanning telescope. The retrieval method is based on the Fraunhofer line method (Grossmann et al., 2018). The field of view (FOV) is 0.7°. At US-NR1 a typical measurement included a scan from nadir to the horizon in 0.7° steps at two

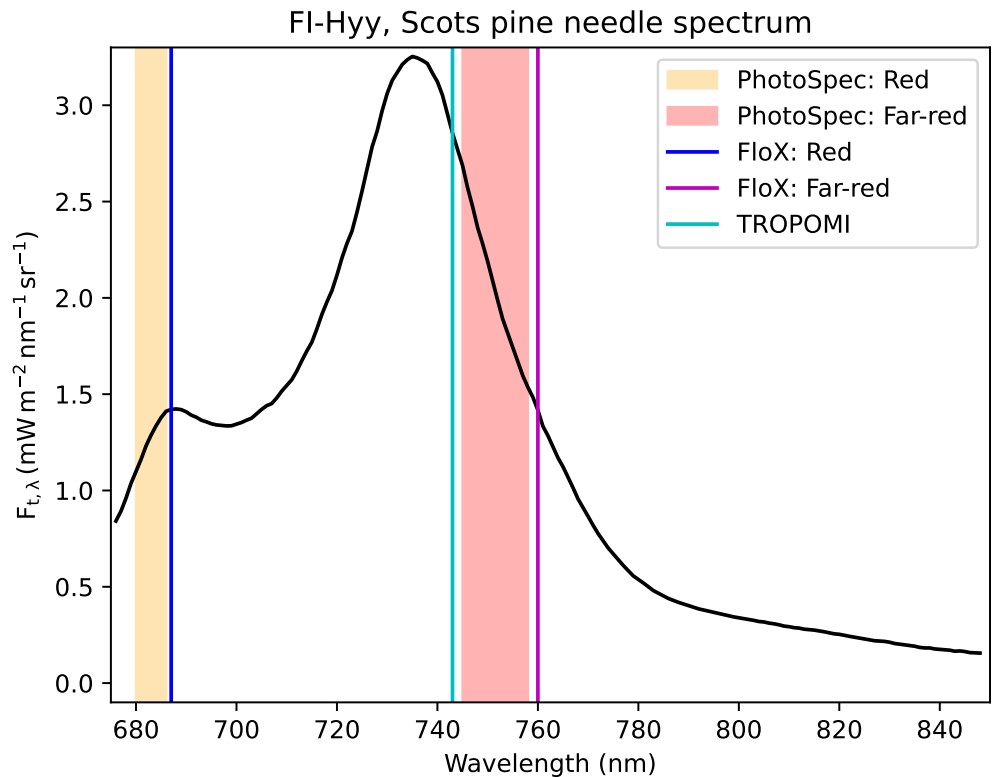

**Figure 1.** SIF emission spectrum for Scots pine located in the southern boreal zone (from Magney et al. (2019b)) with wavelength regions of the observations. The lines indicate the bands for which the SIF signal was retrieved for the FloX observations and TROPOMI and the shaded regions indicate the wavelength regions for which the SIF with the PhotoSpec was retrieved.

different azimuth direction (Magney et al., 2019a). At CA-Obs three vertical scans at three different directions (35°W, 0°N, 35°E) were done in sequence (Pierrat et al., 2021). More details of these observations can be found in Magney et al. (2019a) (US-NR1) and Pierrat et al. (2021) (CA-Obs). At US-NR1, observations were available from June 2017 until June 2018. At CA-Obs we used observations for whole years 2019 and 2020.

In Sodankylä the observations were made with a FloX box (JB Hyperspectral Devices, Düsseldorf, Germany) (https://www.jb-hyperspectral.com/products/flox/). The FloX box observations were used to retrieve SIF in the $O_2B$ band at 687 nm in the red region and the $O_2A$ band at 760 nm in the far-red region. The retrieval method used to process the data was the improved Fraunhofer line method (Alonso et al., 2008; Cendrero-Mateo et al., 2019). We used close to nadir observations from June 2021 until the end of 2021 in our study. The FOV is 25°. The different wavelengths of the retrieved SIF signals by the instruments are

shown in Fig. 1, with the SIF spectrum from observations of Scots pine needles in Hyytiälä, southern Finland (Magney et al., 2019b).





The $CO_2$ fluxes were measured above the canopy. At FI-Sod, measurements were made at 25 m with a Gill HS-50 sonic anemometer (Gill Instruments, Lymington, UK) and a LiCor LI-7200 gas analyzer (LiCor Inc., Lincoln, NE, USA) (Knorr et al., 2025). At CA-Obs the measurement height was 25 m, and the anemometer was CSAT3 (Campbell Scientific Inc., Logan,
UT, USA) and the gas analyzer LI-7200 (Pierrat et al., 2021). At US-NR1 measurements were made at 21.5 m height with the CSAT3 and an LI-6262 gas analyzer (Magney et al., 2019a). Gap–filling and partitioning of the measured net ecosystem exchange flux to gross primary production (GPP) and total ecosystem respiration was done at FI-Sod following Aurela et al. (2015). Flux partitioning and gap–filling at CA-Obs was done as described in Barr et al. (2004) and at US-NR1 using the Reichstein et al. (2005) method with the R package REddyProc (Wutzler et al., 2018).

## 2.2 Remote sensing observations by TROPOMI

The TROPOspheric Monitoring Instrument (TROPOMI) is aboard the Copernicus Sentinel-5P mission and has been providing data since 2018 (Guanter et al., 2021). TROPOMI has global continuous spatial sampling with daily revisit times, because it has a nearly sun-synchronous orbit with a repeat cycle of 16 days and a wide swath of 2600 km (Köhler et al., 2018). The pixel size at nadir was 3.5 x 7.5 $km^2$ at the beginning of the mission and 3.5 x 5.5 $km^2$ after August 2019 (Guanter et al., 2021). The
135 instrument's near infrared band 6 covers the spectral range between 725-775 nm, with a full width at half maximum (FWHM) = 0.38 nm, making it suitable for retrieving far-red SIF (Köhler et al., 2018). We used the TROPOSIF product derived from the 743-758 nm window, at 743 nm (Guanter et al., 2021). The retrieval methodology is based on the Fraunhofer line in-filling principle (Plascyk and Gabriel, 1975) and a data-driven method is used (Guanter et al., 2015). In this study we used 0.5° x 0.5° sampling area around two study sites, CA-Obs and FI-Sod, corresponding to about 56 km x 33 km at CA-Obs. We did
not use TROPOMI data for US-NR1 because TROPOMI observations only covered part of the in situ observational period. TROPOMI's Level 2 cloud fraction product was applied for a strict cloud filtering, removing all SIF data for which the cloud fraction exceeded 0.2.

## 2.3 Model description of QUINCY

The QUantifying Interactions between terrestrial Nutrient CYcles and the climate system (QUINCY) model is a terrestrial
biosphere model (TBM) that can be run on a single site or larger, such as regional or global, scales. Here we give a brief description of the model, further details can be found in Thum et al. (2019).

The complete version of QUINCY has fully coupled carbon, energy, nitrogen and phosphorus cycles. The model has a modular structure that allows only some parts of the model to be run. We used the canopy module, which calculates fast biophysical processes of the model, including stomatal conductance, photosynthesis and radiative transfer within the canopy.
Influence of soil is considered so that water uptake is constrained by soil moisture given a prescribed, plant functional type -specific root profile. Leaf area index (LAI) and leaf nitrogen content are prescribed with a constant value in the canopy module (otherwise these would be calculated prognostically inside the model). The depth (in terms of LAI) of the canopy layers increases exponentially towards the lower canopy layers, of which there are a maximum of ten layers. The nitrogen





gradient decreases with canopy depth according to observations (Niinemets et al., 1998). Leaf stoichiometry, i.e. the nitrogen
to carbon ratio, is fixed in the canopy module.

Photosynthesis is calculated according to Kull and Kruijt (1998). This approach is based on the biochemical model of
Farquhar et al. (1980), but instead of the regular implementation of having the minimum of the two branches limiting photo-
synthesis (light-limited rate of photosynthesis and carboxylation capacity limited rate), the amount of light-saturated region
in the leaf is taken into account. In the non-light-saturated part, photosynthesis is calculated using the light-limited rate of
photosynthesis based on the maximum electron transport rate parameter $J_{max,25}$ (the parameter has been scaled to 25 °C). For
the light-saturated part, photosynthesis is calculated as the minimum of electron transport rate-limited photosynthesis and the
carboxylation capacity limited photosynthesis (determined by the maximum carboxylation capacity parameter $V_{c(max),25}$).

Photosynthesis is calculated separately for sunlit and shaded leaves in each canopy layer and coupled to the stomatal con-
ductance (Medlyn et al., 2011). The fraction of sunlit and shaded leaves are calculated using the radiative transfer scheme
based on the two-stream approach of Spitters (1986), and extended to include canopy albedo, clumping and attenuation of the
shortwave backscatter from the ground. Radiative transfer is calculated separately for the visible and near-infrared bands. Leaf
reflectance is calculated based on the PFT-specific single leaf scattering albedo. Leaf transmissivity is assumed to be equal to
reflectivity. Clumping, non-random distribution of leaf elements, is described according to Campbell and Norman (1998).

The onset of photosynthesis for evergreen coniferous forests is delayed as a function of air temperature in QUINCY (Thum
et al., 2019). The use of a constant temperature response for the photosynthesis parameters $J_{max,25}$ and $V_{c(max),25}$ has been
shown to predict too early spring recovery in boreal coniferous forests (Thum et al., 2008). Therefore, a more accurate rep-
resentation of delayed spring onset of these parameters has been adapted from Mäkelä et al. (2019) and parameterized using
several sites from the FLUXNET database (NOA, 2007). The formulation for the state of acclimation (SOA) ($S$) is (Mäkelä
et al., 2004):

$$\frac{dS}{dt} = \frac{1}{\tau_{soa}}(T_{air} - S) \tag{1}$$

where $\tau_{soa}$ is a time constant (114 hours) and $T_{air}$ is the air temperature. Therefore, $S$ is a delayed temperature sum. It is
used to calculate a delay factor ($\beta_{soa}$) for photosynthesis (used as a multiplier for the parameters $V_{c(max,25)}$ and $J_{max,25}$ as
described in Thum et al. (2019)):

$$\beta_{soa} = \frac{1}{1 + e^{b(S-T_s)}} \tag{2}$$

where $b$ (°C$^{-1}$) and $T_s$ (°C) are parameters, set to -0.5 °C$^{-1}$ and 5.0 °C, respectively.

## 2.4 Leaf level model of chlorophyll fluorescence

We applied the widely used leaf level model for steady-state leaf chlorophyll fluorescence developed by van der Tol et al.
(2014). It is part of the Soil Canopy Observation, Photochemistry and Energy fluxes (SCOPE) model (van der Tol et al., 2009)



and its derivatives, such as mSCOPE (Yang et al., 2017). Here, we briefly introduce the equations of this leaf model to clarify

its implementation in QUINCY. We could not directly follow the implementation in SCOPE, because QUINCY has a different

formulation for the photosynthesis model, which is described in section 2.3. For a detailed description of the model, see van der

Tol et al. (2014).

The ChlF pathway of using excitations in the leaves will be denoted by $F$, photochemistry by $P$. Heat dissipation is divided

into a constitutive thermal dissipation, denoted by $D$, and an energy-dependent heat dissipation, denoted by $N$, and this is NPQ.

$D$ is present in dark-adapted plants, while $N$ is more variable and controlled by the electron transport of the photosystems.

Below we introduce the reversible NPQ and in the next Section 2.4.1 also a formulation for the sustained NPQ, which is an

additive term to the reversible NPQ.

The rate coefficients ($K$) express the probability of different rates of excitation and can be used to express the yield:

$$\Phi_P = \frac{K_P}{\sum K} \tag{3a}$$

$$\Phi_F = \frac{K_F}{\sum K} \tag{3b}$$

$$\Phi_D = \frac{K_D}{\sum K} \tag{3c}$$

$$\Phi_N = \frac{K_N}{\sum K} \tag{3d}$$

$$\sum K = K_P + K_F + K_D + K_N \tag{3e}$$

These rates are mutually exclusive, and therefore the yield of all processes is:

$$\Phi_P + \Phi_F + \Phi_D + \Phi_N = 1 \tag{4}$$

The rate coefficient $K_F$ is constant and $K_D$ depends on the air temperature ($T_{air}$ in °C) as estimated from measurements

by (van der Tol et al., 2014):

$$K_D = MAX(0.8738, 0.0301\frac{1}{C^\circ} * T_{air} + 0.0773) \tag{5}$$

while $K_N$ and $K_P$ are influenced by the metabolic state of the leaves.

According to Genty et al. (1989), the $\Phi_P$ at the steady state can be calculated from the ratio of the variable fluorescence

($F'_m - F_t$) to the maximum fluorescence in light ($F'_m$), as

$$\Phi_P = \frac{F'_m - F_t}{F'_m} \tag{6}$$

where $F_t$ is the steady-state fluorescence. Yields can be expected to follow the Genty relationship:



$$\Phi_{F_t} = (1 - \Phi_P)\Phi_{F'_m} \tag{7}$$

where $\Phi_{F_t}$ is the steady-state fluorescence yield and $\Phi_{F'_m}$ is the yield of maximum fluorescence in light, obtained with a saturating pulse in the PAM observations. To evaluate $\Phi_{F'_m}$, we note that the rate coefficient of photosynthesis $K_P$ goes to zero with the saturating light pulse, since then all the open PSII reaction centers are closed by the pulse. Because of this and Eq. (3c) we get

$$\Phi_{F'_m} = \frac{K_F}{K_F + K_D + K_N} \tag{8}$$

and here only $K_N$ is unknown. van der Tol et al. (2014) developed an experimental relationship to relate $K_N$ to the changes in $\Phi_P$. The $K_N$ that controls $F'_m$ must be related to the relative decrease in the photochemical yield. To achieve this, a factor $x$ is defined which is zero when photochemistry is working at full efficiency and one when photochemistry is completely absent. This is

$$x = 1 - \frac{\Phi_P}{\Phi_P^0} \tag{9}$$

where $\Phi_P^0$ is the maximum photochemical yield which can be observed under dark adapted and low light conditions.

In the original SCOPE formulation, the constraint imposed on $\Phi_P$ is calculated as the fraction of actual electron transport compared to potential electron transport. Since QUINCY has a different formulation for photosynthesis, we calculated the limitation on $\Phi_P$ as a fraction of actual electron transport compared to the case where the whole leaf would be light limited, i.e. there would be no saturated region at all. In principle, our solution is similar, although the formulation of the photosynthesis model is slightly different.

The steady-state chlorophyll fluorescence is

$$\Phi_{F_t} = (1 - \Phi_P^0 + x\Phi_P^0)\Phi_{F'_m} \tag{10}$$

The empirical relationship between $x$ and $K_N$ is (van der Tol et al., 2014)

$$K_N = \frac{(1 + \beta)x^\alpha}{\beta + x^\alpha}K_N^0 \tag{11}$$

where the parameters are $K_N^0 = 2.48$, $\alpha = 2.83$ and $\beta = 0.114$. These are for the standard conditions, while the van der Tol et al. (2014) also gives parameter values for water-limited conditions. This is the reversible part of the $K_N$, which we will refer to as $K_{N_{rev}}$ in the following section.



t

**Table 2.** Variables of leaf level chlorophyll fluorescence model, including values held constant. All of these variables are unitless.

| Variable name (unit) | Symbol | Value |
|---|---|---|
| Rate coefficient for photosynthesis (-) | $K_P$ | - |
| Rate coefficient for fluorescence (-) | $K_F$ | 0.05 |
| Rate coefficient for constitutive heat dissipation (-) | $K_D$ | Eq. (5) |
| Rate coefficient for energy-dependent heat dissipation (-) | $K_N$ | - |
| Yield for photosynthesis (-) | $\Phi_P$ | - |
| Yield for fluorescence (-) | $\Phi_F$ | - |
| Yield for constitutive heat dissipation (-) | $\Phi_D$ | - |
| Yield for energy-dependent heat dissipation (-) | $\Phi_N$ | - |
| Maximum fluorescence in light (-) | $F'_m$ | - |
| Steady-state fluorescence (-) | $F_t$ | - |
| Maximum photochemical yield as observed in dark adapted (-) | $F_P^0$ | - |

The chlorophyll fluorescence yield can be used to calculate the SIF emission per leaf layer $\mathrm{SIF_{cl}}$,

$$SIF_{cl} = Sun_{frac}PPFD_{sun}\Phi_{F_t,sun} + (1 - Sun_{frac})PPFD_{sha}\Phi_{F_t,sha} \tag{12}$$

where $\mathrm{PPFD_{sun}}$ is the absorbed photosynthetically active radiation (PAR) for the sunlit leaves, $\mathrm{PPFD_{sha}}$ for the shaded leaves and $\mathrm{Sun_{frac}}$ is the fraction of the sunlit leaves. The $\Phi_{F_t,sun}$ is the fluorescence yield calculated for the sunlit leaves and $\Phi_{F_t,sha}$ for the shaded leaves. Sections 2.5.1-2.5.3 will describe the different ways in which this emission has been scaled up to the canopy level.

### 2.4.1 Sustained non-photochemical quenching (NPQ$_s$)

Sustained non-photochemical quenching (NPQ$_s$) is a process that is relevant to plants that retain needles through the winter. This is another NPQ mechanism in addition to the reversible NPQ that we introduced in the previous section (Eq. 11). In previous work with the Community Land Model (CLM) model (Raczka et al., 2019), a parameterization for sustained NPQ based on the state of acclimation was developed. We have already used state of acclimation in the photosynthesis model of QUINCY (Eq. 1). Following the earlier work and similarly to the state of acclimation, we obtained for $K_{N_{sus}}$:

$$K_{N_{sus}} = \frac{K_{N_{sus,max}}}{1 + e^{b_{NPQ_s}(S - T_{NPQ_s})}} \tag{13}$$

where $b_{NPQ_s}$ (°C$^{-1}$) and $T_{NPQ_s}$ (°C) are parameters, set to 0.5 °C$^{-1}$ and 5.0 °C, respectively. The difference between this equation and Eq. (2) is that it has large values in winter, while Eq. (2) has large values during the summer. $S$ is obtained from





Eq. (1). To parameterize the Eq. (13) we used SIF observations from US-NR1 and tuned the values to get the best match with our model, since the previous study by Raczka et al. (2019) had developed the parameterization for this site using active leaf

level observations from coniferous forest Hyytiälä in Finland. For the parameters $b_{NPQ_s}$ and $T_{NPQ_s}$ (Eq. 13) we obtained values of 0.5 °C$^{-1}$ and 5.0 °C, respectively. We tested different values and chose those that best fit the data.

When both reversible and sustained NPQ were taken into account, $K_N$ was then a sum of them, as

$$K_N = K_{N_{rev}} + K_{N_{sus}}. \tag{14}$$

### 2.5   Models for the radiative transfer of the SIF signal

### 2.5.1   mSCOPE

The mSCOPE model (Yang et al., 2017) is a further development of the widely used SCOPE model (van der Tol et al., 2009) that has been eventually implemented in SCOPE 2.0 (Yang et al., 2021). In mSCOPE, the canopy is allowed to have a heterogeneous vertical canopy structure, whereas in SCOPE it is assumed to be homogeneous. The QUINCY model has a vertically varying canopy structure, as explained in Section 2.3. Therefore, the use of mSCOPE was more suitable than SCOPE

for coupling with QUINCY.

In mSCOPE, the model Fluspect (Vilfan et al., 2016) calculates leaf reflectance, transmittance and chlorophyll fluorescence. The radiative transfer of mSCOPE is described by two SAIL-based models (Verhoef, 1984): one, which calculates the radiative transfer of incident radiation, and another one, which calculates the radiative transfer of emitted chlorophyll fluorescence. Homogeneity in the horizontal direction is assumed, but heterogeneity of leaf properties in the vertical direction is allowed.

The probability of sunlight on leaves is described by a Poisson model. The shaded leaves are illuminated only by diffuse radiation and their absorbed radiation does not depend on geometry. For the sunlit leaves, the absorbed radiation is calculated for discrete leaf orientations, including 13 leaf inclinations and 36 leaf azimuth angles relative to the solar azimuth. The mSCOPE model calculates the top of the canopy (TOC) value for the ChlF emission.

The mSCOPE model has been implemented in QUINCY. This implementation is now called QUINCY-mSCOPE, replacing

the original QUINCY radiative transfer model (Spitters, 1986). The vertical profile of leaf chlorophyll, that was calculated inside QUINCY, was used to calculate the radiative properties of each layer. mSCOPE runs over 60 canopy layers that were grouped to mimic the usual 10 layers in QUINCY as a function of the QUINCY layer LAI. mSCOPE outputs were then integrated for each layer group to represent each of the 10 QUINCY layers. For stability reasons, we also had to limit the calculation of the radiative transfer code to solar zenith angles below 80°. To test the implementation, we performed a sensitivity

analysis by running simulations with different parameter values in both mSCOPE and QUINCY-mSCOPE. The results were consistent (not shown), so we are confident that there are no major technical errors in the implementation. Using QUINCY-mSCOPE instead of QUINCY-orig caused small differences in the simulated GPP, but overall the results were similar (for CA-Obs the Pearson correlation coefficient ($r^2$) was 0.99 for half-hourly values throughout the time period and the root mean squared error (RMSE) was 0.77 $\mu\text{mol}\,\text{m}^{-2}\,\text{s}^{-1}$). The viewing angle was set to nadir in the QUINCY-mSCOPE simulations.



### 2.5.2 Layered two-stream model (L2SM)

The Layered canopy two-Stream Model (L2SM) (Knorr et al., 2025; Quaife, 2025) is a two-stream radiative transfer model based on the solutions provided by Meador and Weaver (1980) and allows the calculation of diffuse emissions originating from plant leaves. The total emission $E_{canopy}$ leaving the canopy, with the canopy including $L$ layers, is

$$E_{canopy} = \sum_{l=1}^{L} \frac{SIF_{leaf}}{2} \left[ \frac{T_{l-\frac{1}{2}}(1 + R_{L-l-\frac{1}{2}})}{1 - R_{l-\frac{1}{2}} R_{L-l-\frac{1}{2}}} \right] \tag{15}$$

where $R_{l-\frac{1}{2}}$ is the reflectance of the entire canopy above the middle layer $l$ and $T_{l-\frac{1}{2}}$ is the transmittance of the same. $R_{L-l-\frac{1}{2}}$ is the reflectance of everything below the middle of layer $l$, including the soil. $SIF_{leaf}$ is the SIF emission from the layer, and in the formulation used here it is assumed to be equal in the upward and downward directions. The reflectance and transmittance of individual layers are combined using the technique of adding, so the combined reflectance ($R+$) of two layers is given by

$$R_+ = R_1 + T_1^2 R_2 R_M \tag{16}$$

where $R_1$ is the reflectance of diffuse radiation from the upper layer, $T_1$ is the transmittance due to diffuse radiation of the upper layer and $R_2$ is the reflectance due to diffuse radiation from the lower layer (which could also be the soil). $R_M$ takes into account multiple diffuse reflections between layers and is given by

$$R_M = \frac{1}{1 - R_1 R_2}. \tag{17}$$

A detailed derivation and description of the L2SM can be found in Quaife (2025). As a starting point, we calculated the L2SM outside the QUINCY model. We gave the L2SM the leaf reflectance (which also equals transmissivity in QUINCY), the leaf area index for each layer, and the SIF emission per layer (as calculated by Eq. 12).

The formulation of L2SM is a two-stream model, similar to the original radiative transfer model of QUINCY. However, while QUINCY is based on Spitters (1986), the L2SM approach is based on Meador and Weaver (1980). Therefore, the radiative transfer of incoming radiation used to calculate photosynthesis differs slightly from the way SIF is transferred within the canopy.

An addition to earlier implementation of L2SM (e.g Knorr et al. (2025)) was that the attenuation of the SIF signal inside the leaf was taken into account. In this approach the leaf was split into two halves with equal optical depth and it was assumed that the SIF emission would originate from between those two layers. The doubling formula for optical media that has been illuminated by a diffuse flux at some certain wavelength gives for leaf reflectance a similar formulation as in Eq. (16). $T_{\frac{1}{2}}$ and $R_{\frac{1}{2}}$ are the transmittance and reflectance of the media split into two halves, following

$$T_{\frac{1}{2}} = \left( T(1 - R_{\frac{1}{2}}^2) \right)^{\frac{1}{2}}, \tag{18}$$




and:

$$R_{\frac{1}{2}} = \frac{R}{1+T}. \tag{19}$$

When leaf spectra for $R$ and $T$ are given, an emission factor for an emitting factor $E_{leaf}$ for an emitting layer in the middle of a leaf can be calculated as:

$$E_{leaf} = \frac{T_{\frac{1}{2}}}{1 - R_{\frac{1}{2}}^2}. \tag{20}$$

This emission factor is used to calculate leaf-level SIF ($SIF_{leaf}$) by:

$$SIF_{leaf} = E \times SIF_{internal} \tag{21}$$

where $SIF_{internal}$ is the internal emission inside the leaf. For taking into account the wavelength, we used mathematically constructed estimate of the *in vivo* leaf spectrum, which was based on double Gaussian curve and normalized to one.

### 2.5.3   Liu and Zeng approaches (LZ)

In addition to modelling the transfer of the SIF signal, we have also tested some simpler formulations to estimate the SIF leaving the canopy. A more empirical approach was based on the work of Liu et al. (2020) (for the visible region) and Zeng

et al. (2019) (for the near-infrared/far-red region). We used the formulation presented by Hao et al. (2021) for the escape fraction. The escape fraction describes how much of the emitted SIF signal reaches the top of the canopy. The total canopy SIF can be expressed using the escape fraction $f_{esc}$ as (Sun et al., 2017):

$$SIF = PAR \cdot fAPAR \cdot \Phi_{F_t} \cdot f_{esc} \tag{22}$$

where SIF is the observed SIF, PAR is the photosynthetically active radiation, fAPAR is the fraction of absorbed PAR and

$\Phi_{F_t}$ is the chlorophyll fluorescence yield. An empirically based formulation for the escape fraction $f_{esc}$ is:

$$f_{esc}^{reg} = \frac{\rho^{reg}}{fAPAR \cdot \sigma^{reg}} \tag{23}$$

where reg is either visible (vis) or near-infrared (nir) region, $\rho$ is the reflectance of green vegetation and $\sigma$ is the single leaf scattering albedo, which is the sum of reflectance and transmittance. To estimate the escape fraction from QUINCY in this way, we used the modelled vegetation reflectance, fAPAR and the single-leaf scattering albedo constants. After calculating the

escape fraction, the upscaled SIF emission (calculated as the sum of canopy layer SIF emissions from Eq. (12) which were multiplied by the canopy LAI) was multiplied by to obtain the estimate of the SIF signal.



### 2.5.4 Upscaling only

In order to estimate the importance of the escape factors for the different approaches and to see the influence of the different radiative transfer approaches, we also calculated the emission of the SIF signal with a simple upscaling of the simulated leaf level SIF, i.e., without any attenuation of the signal taking place within the canopy and assuming the escape factor then to be one. This approach was therefore integrating emission from all leaves. We calculated the upscaled signal for the red region only, using the leaf level SIF emission from Eq. (12). We summed the different layers of the SIF emission by multiplying each layer by its LAI value.

We did this to have a point of comparison with the other methods and to estimate the gain of the more computationally intensive approaches. We expected this method to give a large overestimation of the SIF signal at the top of the canopy, but we were interested to see if other methods showed superior performance, e.g. in terms of seasonality or diurnal cycle.

### 2.6 Converting the units of SIF from modelled to observed

The output of the radiative transfer approaches in Sections 2.5.2-2.5.4 is in flux units, i.e. $\mu\mathrm{mol\,m^{-2}\,s^{-1}}$. To be able to compare the model output with the observations, which are typically in units of $\mathrm{Wm^{-2}s^{-1}nm^{-1}sr^{-1}}$, we need to convert the units of the model output. This procedure was similar as described in Knorr et al. (2025). The Planck equation was used to obtain energy of photons per mole. For the LZ and Upscaled approaches we used a spectrum observed at Hyytiälä Scots pine forest, located in central Finland (Magney et al., 2019a) (Fig. 1), to weight the relative strength of the emissions at certain wavelengths compared to a reference SIF spectrum (for the L2SM approach this was done already in leaf attenuation part, see Section 2.5.2). The emittance of SIF from the top of the canopy was assumed to be isotropic, so the conversion to steradians was done using a constant factor of $\frac{1}{\pi}$.

### 2.7 Modelling protocol

In this work we took advantage of the modular structure of QUINCY and used only the canopy model in our simulations. The LAI and leaf nitrogen content were prescribed so that the average summertime GPP level matched the observations. The meteorological forcing (air temperature, precipitation, atmospheric pressure, vapor pressure deficit, wind speed, short and longwave radiation) to run the model was obtained from the site measurements. In addition, we used atmospheric $CO_2$ concentration and N deposition, but the N cycle was not active in the canopy model simulations. The simulations were performed for the years of the observations. The leaf single scattering albedo used to calculate reflectance for conifers was 0.15 in the visible wavelength region and 0.73 in the near infrared (Otto et al., 2014). The soil albedo was estimated to be 0.15 in the visible and 0.30 in the near infrared. The LAI values have been set to lower values than observations, to make a better match with the observed magnitude of GPP. The LAI was set to $2.5\ \mathrm{m^2\,m^{-2}}$ in CA-Obs, $3.6\ \mathrm{m^2\,m^{-2}}$ in US-NR1 and $1.2\ \mathrm{m^2\,m^{-2}}$ in FI-Sod.





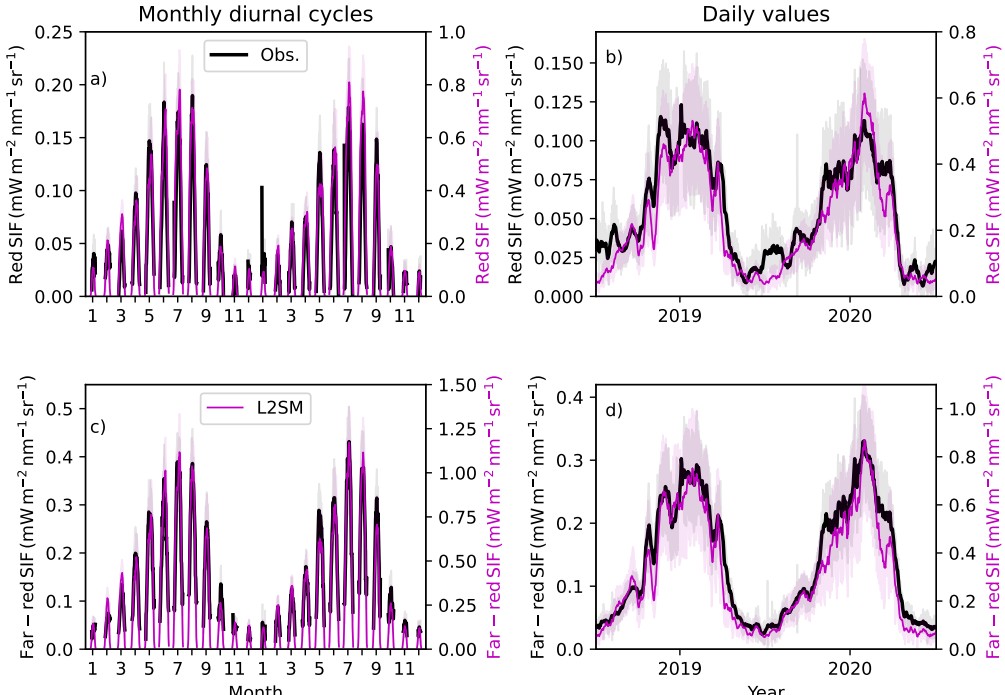

**Figure 2.** Monthly diurnal cycles for red region SIF (a) and far-red SIF (c) and daily values for red region SIF (b) and far-red SIF (d) at CA-Obs. The black line is the observation for all plots, magenta for the L2SM. All the lines for daily values have been smoothed with a 15-day long window. The shaded regions denote standard deviation.

## 3   Results

The diurnal monthly cycles and the seasonal cycles of the observed and simulated SIF signals for the years 2019 and 2020 at CA-Obs are shown in Fig. 2. This was the final result we obtained after testing for different radiative transfer schemes and adding the description for sustained NPQ. In the following sections we describe how we arrived at these results. In the main text we concentrate on results from the CA-Obs site from which we had most data and show in the supplement the results for the other two sites.

### 3.1   Performance of the radiative transfer models

The different radiative transfer schemes were all tested at all three sites. The monthly diurnal cycles and midday values (10 a.m. to 1:30 p.m.) for GPP, red region SIF and far-red SIF at CA-Obs are shown in Fig. 3. The midday values are shown here





to give a better insight into how the magnitude of the variables changes, thus removing the strong influence of the change in day length on the results. The forest in CA-Obs experienced a strong seasonal cycle with low levels of photosynthetic activity between October and March. The highest observed levels of photosynthesis and SIF occurred in July and August. QUINCY was able to capture this seasonal behaviour at the site. In spring 2019 the eddy covariance observations were out of commission and until mid-2019 the GPP data was based on gap–filling.

All of the different SIF transfer schemes overestimated the magnitude of the observed SIF signals (Fig. 3). We decided not to plot the upscaled value in this figure, as its magnitude was much larger than from the other approaches. The large overestimation of the simulation results will be discussed also later in this paper. The $r^2$ and RMSE values for the different methods for the daily means and different times of the day at CA-Obs are shown for SIF and GPP in Table 3 and the simulation results versus observations for the midday values are shown in Fig. 4 with the months marked in different colors.

Without satisfactory modelling performance of GPP, the simulation of the SIF would not be successful, as the photosynthesis model was used in our approach to calculate the leaf level emissions of SIF and this is why we also show evaluation of GPP. The model performance of GPP in CA-Obs was generally good. The $r^2$ between predicted and measured daily GPP was 0.88 (Table 3). The simulation of GPP was best at midday, and slightly less in morning hours. Since there was a long gap in observed GPP and we instead used gap–filled GPP, we also checked the $r^2$ of daily GPP values for the two years separately. For 2019

the $r^2$ was 0.89 and for 2020 0.86, possibly reflecting the fact that simulations did not capture the turbulent nature of eddy covariance observations, but were potentially closer to gap–filled values that estimate the average behaviour of the ecosystem.

The daily $r^2$ values for SIF were close to those of GPP, even slightly higher for the far-red region, which can be considered surprising, because modelling of SIF is based on modelling of GPP (Table 3). Overall, the modelling approaches for SIF were more successful in the far-red region than in the red region. The $r^2$ values in both wavelength regions were better in the

390 midday and afternoon than in the morning. The lower performance in the morning could be due to sun-view geometry and the 3D structure of the canopy casting shades in the spectroradiometer measurement footprint that were not reproduced by the 1D radiative transfer models. When investigating whether the different months showed clearly different patterns in model behaviour (Fig. 4), it was seen that the highest simulated midday values in summer were higher than the linear fit between observations and simulations would imply, suggesting that the model had a tendency to overestimate these values relative to

395 other time periods. The relationship between observed and simulated values appeared to be more curvilinear in the far-red than the red, with some generally lower values simulated for the observation between 0.3 and 0.5 $\mathrm{Wm^{-2}s^{-1}nm^{-1}sr^{-1}}$.

The performance of the different modelling approaches was quite comparable when looking at the $r^2$ values (Table 3). The RMSE values showed greater variation, but due to the large overestimation by all the models, it is probably not useful to use this as a metric to evaluate the model performance. The upscaled value, which did not include any signal attenuation, performed at a

400 comparable level to more sophisticated approaches if only $r^2$ is considered. Of the three different radiative transfer approaches, L2SM appeared to have a consistent level of performance in both wavelength ranges and different times of the day. Overall, the $r^2$ values were quite similar between the approaches, showing that the different approaches did not have a pronounced influence on the temporal patterns of the simulated SIF.



**Figure 3.** Monthly diurnal cycles for GPP (a), red region SIF (c) and far-red SIF (e) and midday values, calculated from winter time between 10 a.m. and 1:30 p.m., for GPP (b), red region SIF (d) and far-red SIF (f) in CA-Obs. The black line is the observation in all plots, the pink line in the GPP plots is the QUINCY simulation. For the SIF plots the red line is the mSCOPE result, magenta the L2SM and blue the LZ approach. All the lines for midday values have been smoothed with a 15-day long window.





**Table 3.** The $r^2$ and RMSE values of simulated versus observed SIF values in the red and far-red regions in 2019-2020, according to different radiative transfer approaches at CA-Obs. The metrics are also shown for the GPP derived from the standard QUINCY configuration. The morning values are from 6 a.m. to 9:30 a.m., the midday values from 10 a.m. to 1:30 p.m., and the afternoon values from 2 p.m. to 5:30 p.m.

| Variable (unit) / $r^2$ (RMSE) | Daily | Morning | Midday | Afternoon |
|---|---|---|---|---|
| GPP ($\mu$mol m$^{-2}$ s$^{-1}$) | 0.88 (1.47) | 0.83 (2.71) | 0.87 (2.19) | 0.84 (2.15) |
| Red region SIF (Wm$^{-2}$s$^{-1}$nm$^{-1}$sr$^{-1}$) | | | | |
| mSCOPE | 0.85 (0.21) | 0.68 (0.23) | 0.88 (0.29) | 0.82 (0.32) |
| L2SM | 0.84 (0.27) | 0.68 (0.27) | 0.86 (0.31) | 0.84 (0.34) |
| LZ | 0.83 (0.59) | 0.68 (0.54) | 0.86 (0.66) | 0.83 (0.72) |
| Upscaled | 0.84 (1.45) | 0.68 (1.34) | 0.88 (1.89) | 0.84 (1.70) |
| Far-red region SIF (Wm$^{-2}$s$^{-1}$nm$^{-1}$sr$^{-1}$) | | | | |
| mSCOPE | 0.92 (0.18) | 0.87 (0.21) | 0.91 (0.27) | 0.89 (0.28) |
| L2SM | 0.91 (0.31) | 0.87 (0.31) | 0.90 (0.38) | 0.92 (0.38) |
| LZ | 0.91 (1.20) | 0.87 (1.13) | 0.90 (1.49) | 0.92 (1.43) |

Running the same simulations at other sites allowed further evaluation of model performance and possible influences of

405 instrumentation on the diurnal dynamics. For the US-NR1 results we used the same parameterization for sustained NPQ as for CA-Obs. For the FI-Sod we did not include sustained NPQ. We discuss this further in Section 3.2.

At US-NR1, measurements were available from June 2017 to early June 2018. US-NR1 is at a lower latitude than the other study sites, but the high elevation conditions result in a pronounced seasonal cycle with seasonally below freezing winters. The forest is photosynthetically active from May to September, with the shoulder season to winter occurring in October and

410 December, and spring recovery occurring in April and May (Fig. S1).

The model performance for simulating GPP was lower in US-NR1 than in CA-Obs ($r^2$ for daily values was 0.80, Table S1, Fig. S1). The $r^2$ values of the different SIF simulations were much lower in US-NR1 than in CA-Obs (Table S1). This is clearly seen in the amount of scatter between simulated and observed values in Fig. S2. The model seemed to have difficulty in capturing the variation in midday values during the summer months. The seasonal cycle of SIF in US-NR1 was not as well

reproduced as in CA-Obs (Fig. S3b vs. Fig. 5b), although the spring recovery of GPP seemed to be well simulated (Fig. S1b).

Similar to CA-Obs, the $r^2$ values for SIF were generally higher in the far-red region than in the red region at US-NR1(Table S1). The model performance was best at midday. The comparison between the different radiative transfer approaches showed that the upscaled value without any described radiative transfer had the highest $r^2$ values in the red region, and mSCOPE in the far-red region.

FI-Sod is located 100 km north of the Arctic Circle, so winter radiation is zero and temperatures are low. Spring recovery occurs in April and May, the photosynthetically active period is from June to August and photosynthesis ceases in September



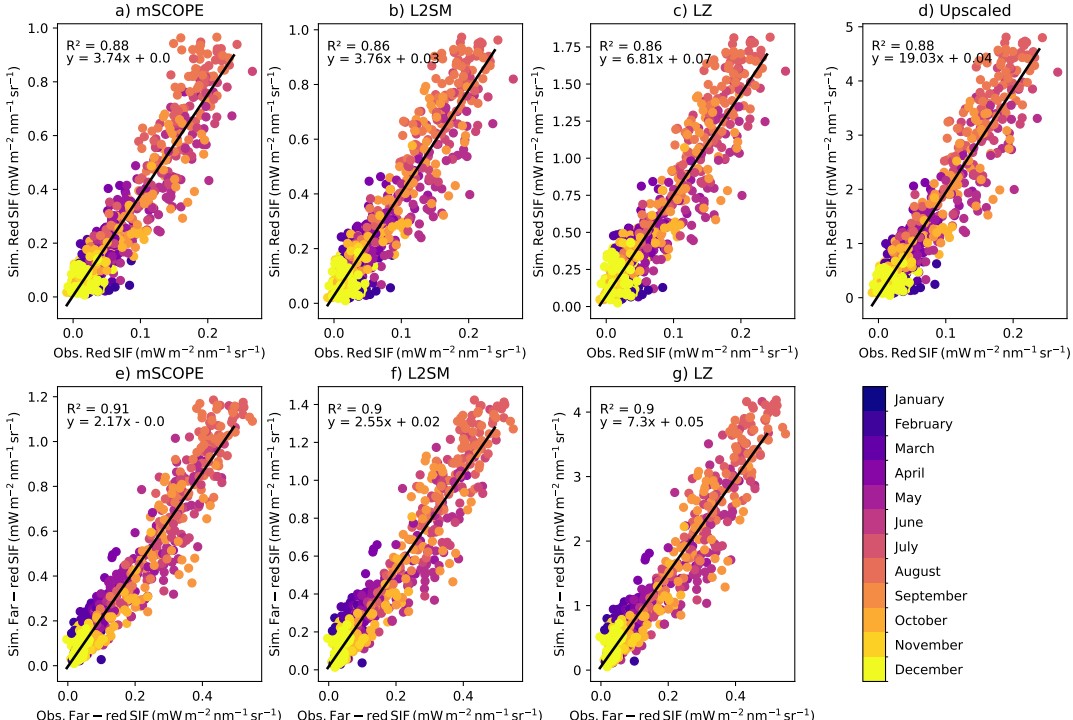

**Figure 4.** Observed vs. modelled SIF midday values in the red region (QUINCY-mSCOPE: a, Q-L2SM: b, Q-LZ: c, upscaled: d) and far-red region (QUINCY-mSCOPE: e, Q-L2SM: f, Q-LZ: g) at CA-Obs. Values from different months are color-coded. The black line shows a fit with the corresponding parameters shown in each panel.

and October (Fig. S4). QUINCY was more successful in simulating GPP at FI-Sod than at the other two sites (Table S2). Modelling SIF was less successful than modelling of GPP in FI-Sod (Table S2, Fig. S5). In both spectral regions, the model performed best in the morning and midday, and worse in the afternoon. Differences between the radiative transfer approaches

were not pronounced.

The magnitude of the observed SIF was similar in both the red and far-red regions at the two sites with the PhotoSpec observations (Table S3) for the July-August midday values. Compared to these values, the FI-Sod value observed with FloX was higher in the red region and lower in the far-red region, which was consistent with what we see in the spectral shape of the SIF emission (Fig. 1). In the far-red region this difference was more pronounced and was half of the value observed with

430 PhotoSpec (Table S3). The overestimation of SIF by the different radiative transfer methods was most pronounced for the sites with PhotoSpec observations in the red region. In the far-red region, mSCOPE had the lowest overestimation and LZ approach the highest.

The escape fraction was calculated as a fraction of the averaged simulated midday SIF value in July-August and the corresponding upscaled value. The results were around 0.2 in the red region, with some higher values with the LZ approach (Table





**Table 4.** Escape fractions for different radiative transfer approaches calculated for July-August midday values (10 a.m. to 1:30 p.m) as a fraction of the upscaled value for the three sites.

| $f_{esc}$ | CA-Obs | US-NR1 | FI-Sod |
|---|---|---|---|
| Red region SIF | | | |
| mSCOPE | 0.19 | 0.17 | 0.21 |
| L2SM | 0.20 | 0.16 | 0.26 |
| LZ | 0.37 | 0.33 | 0.52 |
| Far-red region SIF | | | |
| mSCOPE | 0.25 | 0.24 | 0.17 |
| L2SM | 0.30 | 0.26 | 0.24 |
| LZ | 0.85 | 0.75 | 0.73 |

4). In the far-red region there was more variation between the estimated escape fractions. Generally the escape ratios were lower in the red region compared to the far-red, which was what would be expected. mSCOPE predicted the lowest values of the approaches in the far-red. As the differences in performance of mSCOPE and L2SM radiative transfer approaches were quite similar, the remainder of the analysis is mainly based on the use of the L2SM, as it was computationally efficient to run.

## 3.2 Importance and generality of sustained NPQ modelling

SIF modelled without sustained NPQ showed a strong relationship with the absorbed PAR (aPAR) at CA-Obs (Fig. 5a). A CA-Obs simulation was carried out to assess the performance of the parameterization carried out at US-NR1. QUINCY was successful in simulating the aPAR at CA-Obs at midday (Fig. 5a). The seasonal cycle was strong with winter values of aPAR around 200 $\mu mol\, m^{-2}\, s^{-1}$. The increase towards summer values in aPAR started earlier than the ecosystem response, as the low temperatures prevented the spring recovery of vegetation. The increase towards summer values of aPAR started already 445 in the first part of the year, much earlier than the SIF values started to increase strongly (Fig. 5). The simulated SIF without the described $NPQ_s$ followed the behaviour of the absorbed PAR. The simulated SIF with the $NPQ_s$ was more similar to the observed seasonal behaviour (Fig. 5b). The magnitudes between the simulations and observations differed for SIF, but the general timing was better for the simulation with $NPQ_s$. The $NPQ_s$ had a strong influence on the chlorophyll fluorescence yield ($\Phi_F$) in the model (Fig. 5c). There was no feedback from the modelling of SIF back to the photosynthesis part of the 450 QUINCY model currently, so describing the $NPQ_s$ did not influence modelling of GPP. However, similar mechanism based on state of acclimation as used in $NPQ_s$ has been implemented for spring recovery of GPP in the QUINCY model already earlier.

As US-NR1 is more southerly than CA-Obs, the absorbed PAR did not show a pronounced seasonal cycle (Fig. S3a). This was reflected in the simulated SIF, so that the simulation without $NPQ_s$ showed no clear seasonal cycle. Using the formulation



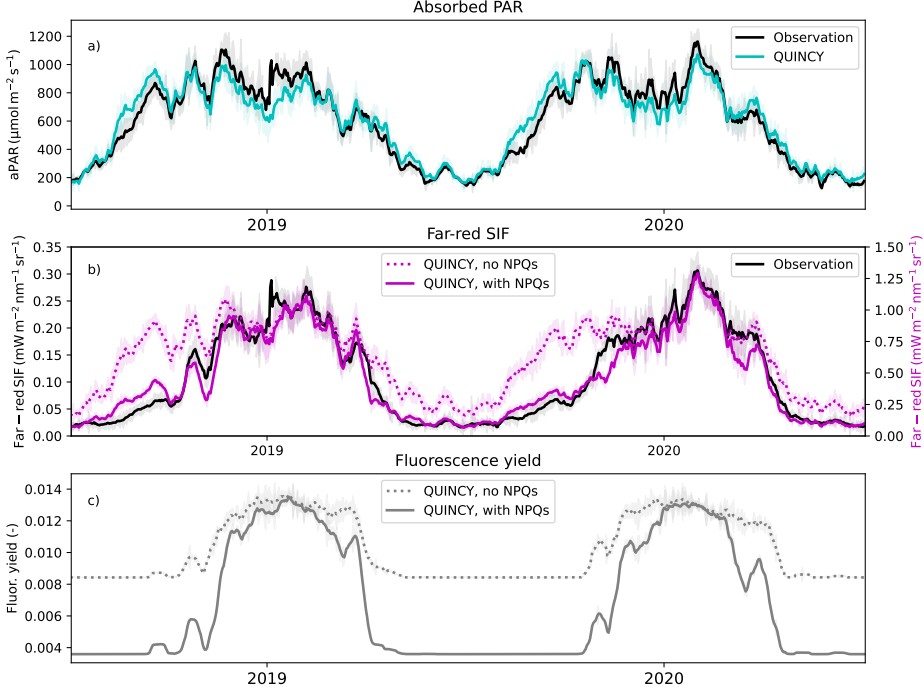

**Figure 5.** The observed and simulated absorbed radiation at CA-Obs (a), near-infrared SIF values with and without sustained NPQ simulated with L2SM (b) and simulated chlorophyll fluorescence yields with and without sustained NPQ (c). Values are averages of midday values (10 a.m. to 1:30 p.m.), the standard deviation is shown as shaded areas. All the lines for midday values were smoothed with a 15-day long window.

for $NPQ_s$ gave the simulated SIF a seasonal cycle, but the formulation used delayed the spring recovery in 2018 too much. The formulated $NPQ_s$ also slightly affected the summer time values, which is physiologically unlikely to happen in reality.

As FI-Sod is located north of the Arctic Circle and the absorbed PAR had a pronounced seasonal cycle (Fig. S6a). The colder air temperatures caused the $\Phi_F$ to drop to a lower level than at the other two sites (Fig. S6b). However, even with this decrease, the absorbed PAR caused the simulated SIF values to start to increase already in February, well before the start of the vegetation active period. Therefore, it seems that the same parameterization that gave reasonably satisfactory results at the other two sites was not as functional at this more northern site with different climatic characteristics (Fig. S6b).

## 3.3 Dependencies between far-red SIF, GPP and PAR

Noticeable differences were found comparing the relationship between GPP and far-red SIF in the observations and simulations with L2SM for June and July (Fig. 6). The observations presented equally high far-red SIF values for CA-Obs and US-NR1, although the observed GPP values were higher at CA-Obs (Fig. 6a). Both observed GPP and far-red SIF values were lower



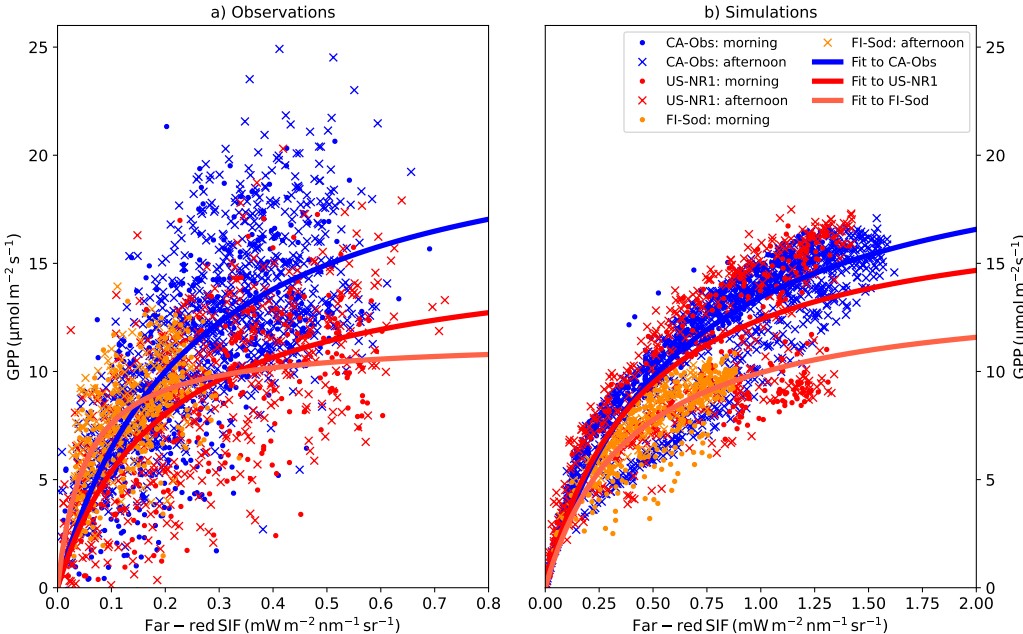

**Figure 6.** The observed (a) and simulated (b) GPP vs. far-red region SIF relationship at three different sites for half-hourly values for all points in June and July using the L2SM approach in the simulations. CA-Obs has half-hourly values and FI-Sod and US-NR1 hourly values. The values before noon are denoted as dots, the values after noon are denoted as crosses. Hyperbolic fits are shown as solid lines in the figure.

at FI-Sod compared to the North American sites (Fig. 6a). At FI-Sod the instrument measuring SIF was different and far-red SIF retrieved at different wavelengths. Overall, the observed GPP values featured high variability for simultaneous SIF observations. The observations did not show clear differences in the far-red SIF-GPP relationship between the morning and the afternoon values. The simulated values showed much less scatter for the SIF-GPP relationship than the observations (Fig. 6b). The highest far-red SIF values were obtained at CA-Obs, although the highest GPP values were obtained at US-NR1. Some of

the afternoon values were at a lower level than the morning values, this was seen especially for the CA-Obs simulation results. When looking at the GPP and far-red SIF light responses (Fig. 7 e, g) it was noticed that the simulated far-red SIF values were highly correlated with the PAR values, while simulated GPP values had more scatter. This was caused by the stomatal conductance lowering the GPP and it seems that this variation did not get reflected to simulated far-red SIF values.

    We made a hyperbolic fit to these relationships, fitting parameters $a$ and $b$ of function y=ax/(b+x) (Damm et al., 2015; Pierrat

et al., 2022b). The fitted lines are shown in Fig. 6 and the fitted parameter values with their associated uncertainties are shown in Table S4. The simulated values had generally smaller uncertainties for the fitted parameter $a$, demonstrating the curvilinear relationship between the variables in the simulations that was not as pronounced in the observations, which can be partly be contributed to the high variability in the observations. However, for FI-Sod the small parameter $b$ value in fit to observations (Fig. 6b) leading to a strong curvature in the fitted curve also shows that for the FI-Sod results this relationship is quite linear.





**Table 5.** Model performance in terms of $r^2$ and RMSE for GPP and SIF in the far-red region at three sites for half-hourly values during five summer days and the averaged diurnal cycle over the five days. Calculated for half-hourly values at CA-Obs and hourly values at US-NR1 and FI-Sod.

|  | CA-Obs | US-NR1 | FI-Sod |
| --- | --- | --- | --- |
| GPP, $r^2$ | | | |
| half-hourly | 0.60 | 0.26 | 0.98 |
| averaged | 0.80 | 0.51 | 0.99 |
| GPP, RMSE ($\mathrm{\mu mol\,m^{-2}\,s^{-1}}$) | | | |
| half-hourly | 3.42 | 2.77 | 0.59 |
| averaged | 2.08 | 1.79 | 0.50 |
| SIF, $r^2$ | | | |
| half-hourly | 0.69 | 0.29 | 0.40 |
| averaged | 0.83 | 0.66 | 0.50 |
| SIF, RMSE ($\mathrm{Wm^{-2}s^{-1}nm^{-1}sr^{-1}}$) | | | |
| half-hourly | 0.57 | 0.52 | 0.54 |
| averaged | 0.51 | 0.46 | 0.53 |

Given the worse model performance for SIF at US-NR1, the diurnal cycle during the summer was examined in more detail for all three sites. First, we calculated the $r^2$ and RMSE values for the instantaneous values over five days versus the averaged diurnal cycle over five days. The improvement in $r^2$ and RMSE when moving from instantaneous to averaged values was considerable (Table 5).

The $r^2$ values of the averaged diurnal cycle were comparable for GPP and SIF in the far-red region at CA-Obs (Table 7).
On day of year (DOY) 187, the simulated GPP showed an almost sinusoidal diurnal behaviour, which resulted from simulation under high irradiance conditions (Fig. 7a). The observed GPP showed more variation during the day, which was maybe due to the canopy shading, the role of understorey vegetation and turbulence conditions. The light response of the observed GPP was much more scattered than that of the simulations (Fig. 7e). The simulated far-red SIF values with all the approaches were able to capture variations quite successfully on DOY 189 (Fig. 7c), when variations in radiation occurred.

At US-NR1 the model performance remained significantly lower for GPP than at the other two sites (Table 5) even after calculating the average over five days. The light response curves of the observed GPP and SIF in the far-red region were quite scattered at this site (Fig. S7e, f). The model performed very well for GPP at Sodankylä (Table 5, Fig. S8). The modelled diurnal cycle of far-red SIF did not capture the variation in the observations (Fig. S8c, d). The footprint of the FloX might be partially shadowed during some time periods in sunny days, causing mismatches between the observed and simulated far-red





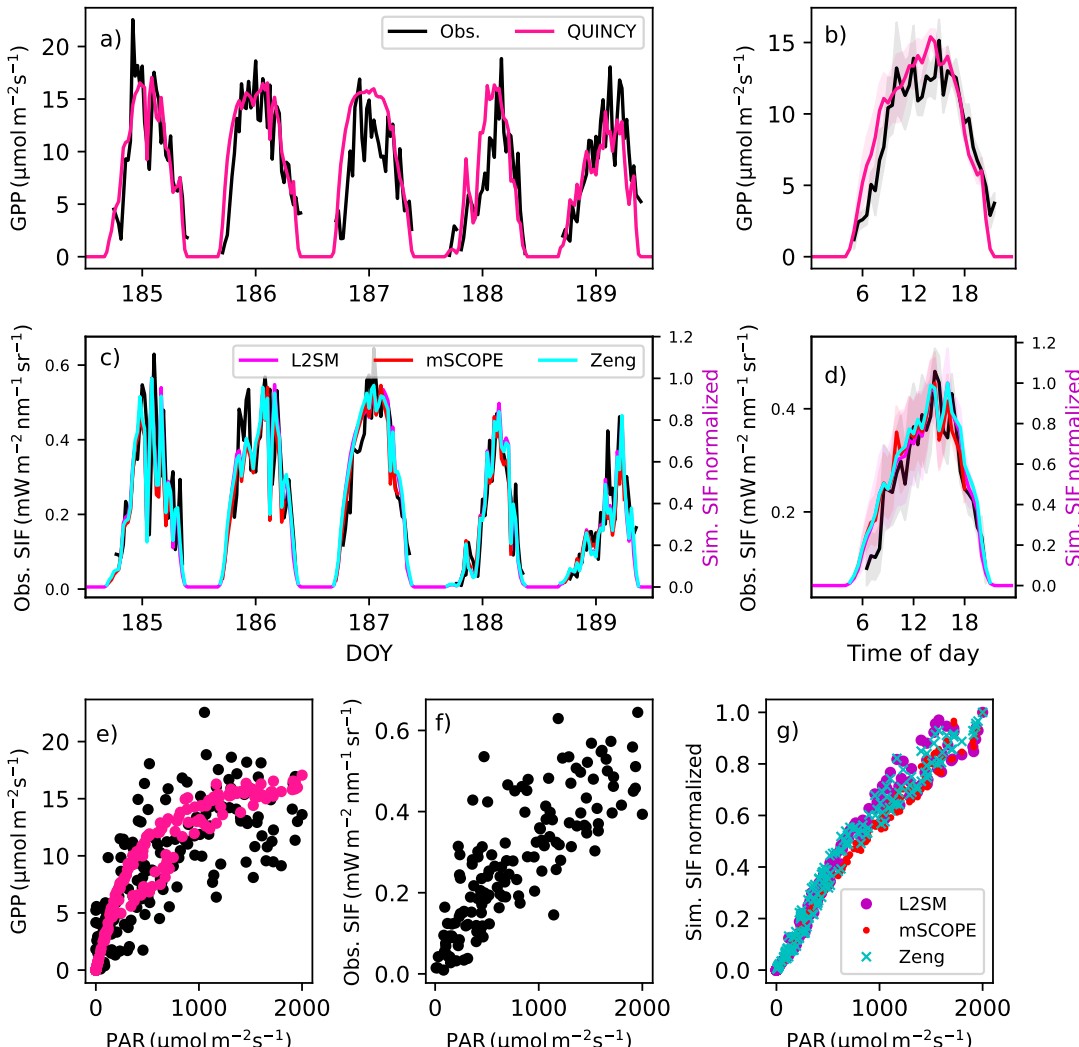

**Figure 7.** The observed and simulated GPP (a) and far-red region SIF (c) for days 185-189 (5 July-8 July 2020) and averaged over these five days (b for GPP and d for far-red region SIF) at CA-Obs. The shaded regions in b and d show the standard deviations of the averaged values. The light response of the observed and simulated GPP for these five days (e), the observed far-red region SIF (f) and the simulated far-red region SIF (g). The observations are in black, the simulated GPP is in pink and the simulated far-red SIF from L2SM is in magenta, from mSCOPE in red and from Zeng approach in cyan. The simulated SIF values have been normalized to one.

SIF (Fig. S8c,d). This might partly explain the lower model performance in the afternoon (Table S2), as the shadow effects might become larger in the shoulder seasons at a high latitude site. Designing an observation without shadow effects would be very challenging at such high latitudes with very long days during summer.





### 3.4 Use of satellite observations in model development

The magnitude of the simulated SIF was large compared to the tower-based observations at all sites (Figs. 3, S1, S4). Comparison with satellite observations at CA-Obs showed a better agreement between the simulated and observed magnitudes (Fig. 8c) than proximal sensing (Fig. 8b). In 2019, the seasonal cycle of TROPOSIF was smoother than that of the simulated SIF and PhotoSpec observations at the site, reflecting the fact that satellites have a limited capacity to detect rapid phenological changes given the spatial and temporal averaging (Fig. 8b). Also, the cloudy days were filtered out from the TROPOSIF, while these were included in the in situ observations, a fact that further smoothed the satellite observations. In 2020, the seasonal cycle was smoother in the PhotoSpec observations and the simulated seasonal cycle was more consistent with TROPOSIF. The simulation results and the TROPOSIF are shown on different scales because the magnitude of the winter TROPOSIF observations is below zero due to the retrieval method, but the minimum of the simulated SIF is zero and using different scales helps to see the seasonality of the two observations together.

The mean daily value with standard deviation for the TROPOSIF at CA-Obs for the period July-August was $0.80 \pm 0.24$ $\mathrm{Wm}^{-2}\mathrm{s}^{-1}\mathrm{nm}^{-1}\mathrm{sr}^{-1}$. The simulations for this time period overestimated the value by 30 %. At the FI-Sod site, both the TROPOSIF product and simulations gave lower values than at CA-Obs. The TROPOSIF estimate for the July-August period was $0.60 \pm 0.24$ $\mathrm{Wm}^{-2}\mathrm{s}^{-1}\mathrm{nm}^{-1}\mathrm{sr}^{-1}$, which the simulations overestimated by 45 %. The values of TROPOSIF were significantly higher for CA-Obs than for FI-Sod, by 33 %. This reflects the difference in GPP observations. The midday GPP at CA-Obs was on average 63 % higher than at FI-Sod.

## 4 Discussion

### 4.1 On the choice of radiative transfer approach

We first tested different ways of describing the radiative transfer of the SIF to decide which method would be robust enough for feasible calculations in a large-scale model. In general, all different methods greatly overestimated the magnitude of the in situ observations by a factor of many (Table S3), making it difficult to use the magnitude as a metric to judge model behaviour.

The overestimation of SIF in evergreen conifer forests also occurred in other modelling studies (Li et al., 2022), and our results are close to the model average compared to a model comparison study conducted at US-NR1 (Parazoo et al., 2020). Our comparison with the satellite observations showed that the discrepancy was not as pronounced as in the site-level observations. It may therefore be that the characteristics of the in situ sampling in this type of ecosystem provide smaller SIF signals than expected. Satellite observations also have a larger footprint that would in CA-Obs also include larch forests. Also, as the in situ observations were tower-based, it is possible that part of the observation footprint contained bare soil at FI-Sod site, which will hence have a lower SIF value, but our model only simulates one plant functional type per site and is horizontally homogeneous. In reality also the directional effects and shadows influence the observed optical signal (Hilker et al., 2008b, a). The PhotoSpec retrievals were filtered by having an Normalized Difference Vegetation Index (NDVI) based threshold to ensure that only observations of vegetation were used. Preliminary model tests on other ecosystems did not show such large discrepancies





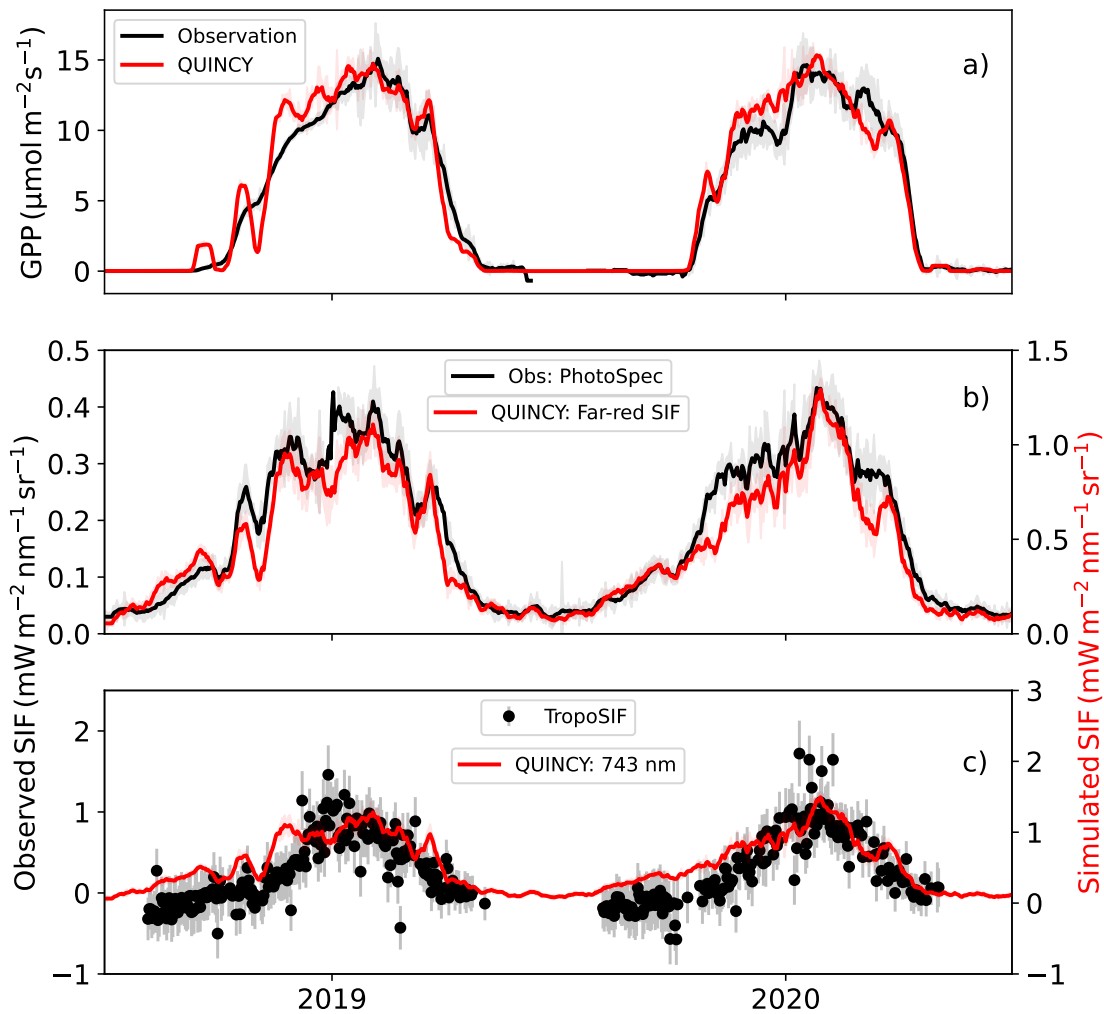

**Figure 8.** The observed and simulated GPP at CA-Obs (a), near-infrared SIF from PhotoSpec observations and simulations with L2SM and (b) and TROPOSIF observations and simulations with L2SM at 743 nm (c). The other values than TROPOSIF are averages of midday (10 a.m. to 1:30 p.m.) values, with standard deviation is shown as shaded areas and TROPOSIF uncertainty shown in error bars (c). TROPOSIF values are daily. All the lines for midday values have been smoothed with a 15-day long window.

between the simulation results and in situ observations (results not shown). The satellite observations exploit also different wavelengths than the tower observations for SIF retrieval, which may play a role in how large the difference is between the observations and simulation results. As we only studied here the satellite observations at two sites, no conclusions can be drawn about how they generally relate to e.g. site level observations.





Overall, when considering the $r^2$ metric for the different approaches, it was found that simple upscaling of the signal did not
perform significantly worse than more sophisticated approaches (Tables 3, S1, S2). This justifies the rather simple approaches
previously used in radiative transfer of SIF (e.g., Lee et al., 2015; Thum et al., 2017), but is in contrast to other studies (Li et al.,
2022). The mSCOPE model performed as well as the other approaches across all sites and often gave best estimates, but its
longer computation time makes it impractical for use in large scale applications (Li et al., 2022). The radiative transfer model
in mSCOPE is based on SAIL, which was originally developed for croplands (van der Tol et al., 2009). The radiative transfer
model in QUINCY was also originally developed for crops. A study by Li et al. (2022) found a significant improvement in
modeling SIF with CLM when they included clumping in their model in a simple manner compared to simulations without
clumping included. Not considering clumping can lead to significant errors in SIF modelling (Zeng et al., 2020). QUINCY has
clumping described in a simple way. There is potentially room for improvement in SIF modelling if clumping is described more
rigorously. The challenge in conifer ecosystems is that the clumping of the canopy exposes more ground vegetation visible to
optical measurements and therefore also affects the remotely sensed signal (Gopalakrishnan et al., 2023).

The structure of the vegetation affects the observed SIF signal (Magney et al., 2019b; Sun et al., 2023a). This is something
that could be addressed by including vegetation NIRv, a near-infrared reflectance that has very similar interactions within the
canopy to the far-red region SIF. Including NIRv, which can be used to approximate the escape ratio, in the analysis would
allow the attribution of the structural effects seen in the SIF signal (Zeng et al., 2019; Dechant et al., 2022). Another approach
would be to account for the 3D structure of the canopy in QUINCY and the SIF RT model. In this way, the GPP and SIF would
both be modulated in a consistent way (Stretton et al., 2025).

L2SM supports vertically varying optical properties, but in this work it has been set up with properties that remain unchanged
through the canopy profile. However, the vertical profile of leaf chlorophyll content could be used to redefine the optical
properties of each layer. In the current version of QUINCY, the chlorophyll content of canopy layers increases with depth. The
single leaf scattering albedo of forest plant functional types in QUINCY is based on a study by Otto et al. (2014). The values
commonly used to calculate reflectance and transmittance in terrestrial biosphere models have been criticized, and some new
approaches based on more data have been proposed (Majasalmi and Bright, 2019). Reflectance was also used in the calculation
of the LZ approach, so it would also influence these results. Also the assumption of having equal reflectance and transmittance
in QUINCY can be argued (Majasalmi and Bright, 2019) and further developed. Also the simplification of using only two
radiative bands (visible and near infrared) might cause some biases for the results in certain wavelengths.

The use of the L2SM and the LZ approaches requires the use of SIF spectra in the unit conversion from the modelled units to
observed units (see Section 2.6). In this work we used for the LZ approach spectra that were measured in a Finnish Scots pine
forest. To extend this approach further to other PFTs, it would be necessary to use other measured SIF spectra. This approach
has limitations in terms of generalizability, as these spectra differ between species and therefore using a single spectrum for
a PFT may introduce uncertainties. Furthermore, the spectral shape of SIF emission changes under stress conditions, e.g. the
photosystems I and II have different responses to stress (Magney et al., 2019a), which will further limit our approach, and
require careful investigation of the stress effects. For the L2SM approach we used an estimate of the in vivo spectrum. This
approach would also benefit from further testing with observed spectra.



The Fraunhofer line method used with PhotoSpec instruments is less susceptible to atmospheric attenuation than using the oxygen lines with FloX. The distance from the soil for FloX at FI-Sod was around 19-20 m and the distance to the canopy less. Therefore the measurement distance is less than 20 m, which is considered the threshold for needing the data to be corrected for atmospheric effects (Sabater et al., 2018; van der Tol et al., 2023). FI-Sod also has a sparse canopy compared to the other sites, which may imply that the footprint of the observing optical fibre is susceptible to environmental influences other than the canopy, such as understory.

The measurements that we used have several uncertainties. The tower SIF observations are relatively new observations and have uncertainties related to instrumentation, the retrieval method, and spatial match of optical and flux footprints (Buman et al., 2022; Cendrero-Mateo et al., 2019; Pacheco-Labrador et al., 2019). Our results showed that using averaging for the model evaluation might be a useful way to use these data. Looking at the whole diurnal cycle showed some discrepancies between the simulation results and the observations, potentially revealing effects of shadows that would require more rigorous modelling tools, such as using 3D description for the forest structure. The fact that the simulated SIF gave similar values for the same GPP value depending on the time of day (Fig. 6) gives reason to look deeper, whether the simulated SIF gives reasonable results in water stressed conditions. However, due to the scattered nature of the SIF observations, it was not straightforward to make a detailed analysis of that. The eddy covariance observations have uncertainties associated with measuring equipment, heterogeneity of the footprint and stochastic nature of turbulence (Richardson et al., 2006). The gap-filling introduces more uncertainties to the data (Vekuri et al., 2025).

## 4.2 Role of sustained NPQ

Our results showed a strong influence of sustained NPQ for these sites, where the plants cannot use the energy of the incoming radiation due to temperature constraints and winter dormancy (see also (Pierrat et al., 2024)). ChlF is closely related to photosynthesis, as both result from the functioning of the leaf biochemical machinery.

In general, simulating NPQ has been a challenge in the modelling community, as it is composed of many processes (Zaks et al., 2013) and active measurements have been needed to quantify it. However, recent advances in the use of spectral imaging of xanthophyll cycle pigments are advancing and making it possible to quantify NPQ also from the optical observations (Pescador-Dionisio et al., 2025; Van Wittenberghe et al., 2024), thus making remote sensing of NPQ feasible. Some new parameterizations for NPQ based on site-level observations have also become available (Martini et al., 2022).

A more thorough analysis on additional data at Sodankylä (including longer time series of the FloX observations and active ChlF observations with MoniPAM) will be carried out to develop a formulation for sustained NPQ applicable to the site. A recent study combining PAM observations worldwide evaluated the photosynthetic capacity of photosystem II and showed that it depends more on the temperature regime of the vegetation's environment than on the plant species (Neri et al., 2024). The study also provided a quantification of this dependence. The final aim will be to obtain a sufficiently general parameterization for the whole coniferous evergreen forest region, using the results of (Neri et al., 2024), possibly with the help of TROPOMI observations. The use of the same model for both photosynthesis and ChlF (Johnson and Berry, 2021) could potentially circumvent this problem. However, the current photosynthesis formulation of QUINCY has a direct influence of nitrogen on the



photosynthesis parameters and also leaf chlorophyll content is closely coupled to photosynthesis (Thum et al., 2025). Leaf chlorophyll content can potentially be useful as a metric related to nitrogen cycling in model evaluation (Miinalainen et al., 2025), which is much needed (Kou-Giesbrecht et al., 2023). In addition, the amount of leaf chlorophyll influences how much of the SIF emitted from the leaves is attenuated in the canopy. Therefore, a model that includes both leaf chlorophyll and SIF included would be beneficial for understanding Earth system processes. The amount of chlorophyll in leaves also affects the shape of the SIF spectrum, and as we used a fixed SIF spectrum to convert from the total flux of SIF to SIF at a given wavelength (Magney et al., 2019b), this would also benefit from some further investigation.

## 5 Conclusions

We have implemented chlorophyll fluorescence into the QUINCY model and tested different canopy transfer approaches of the SIF signal. On a seasonal scale, many of the approaches performed similarly, and did not show clear differences in performance when looking at day-to-day variation and the ability to simulate different times of day. The magnitude of the tower-based SIF observations was greatly overestimated in the simulations, but the timing and seasonality were captured successfully. Of the approaches studied, L2SM showed consistent performance across the sites and is computationally feasible to implement in a large-scale model. Still, the consistent overestimation of SIF suggests the 3D structure of conifer forest should be better accounted for by models and proximal sensing measurements.

Sustained NPQ was relevant in decoupling the simulated SIF from the observed absorbed PAR and the same parameterization improved model performance at the North American sites but appeared less suitable at the Finnish site. The process is likely linked to the air temperature regime of the sites. The satellite TROPOSIF product was able to capture the low spring values observed at CA-Obs and could therefore probably serve as additional data when implementing the parameterization of sustained NPQ in a global model. Use of TROPOSIF observations additionally in model evaluation and development seems feasible.

The next step of this work will be to extend it to other ecosystems, using both in situ and satellite observations as evaluation data. SIF observations have also been shown to be closely linked to the latent heat flux observations and we will extend our analysis also to include these. Together with QUINCY's diagnostic leaf chlorophyll content, a variable which can be observed from space, this work brings QUINCY closer to being a tool for comprehensive analysis of biogeochemical cycles. In addition, the knowledge gained from this work will pave the way for data assimilation studies using SIF observations measured at different scales.

*Code and data availability.* The scientific part of the QUINCY code is available under a GPL v3 license. The scientific code of QUINCY relies on software infrastructure from the MPI-ESM environment, which is subject to the MPI-M License Agreement in its most recent form (https://www.bgc-jena.mpg.de/en/bsi/projects/quincy/software, last access: 23 May 2025). The source code is available online https://doi.org/10.17871/quincy-model-2019), but its access is restricted to registered users. Readers interested in running the model should request a username and password via the Git repository. Model users are strongly encouraged to follow the fair-use policy: https://www.bgc-jena.mpg.de/en/bsi/projects/quincy/software.



L2SM-code by T. Quaife is available at Zenodo in doi: 10.5281/zenodo.13753268. The FloX observations with meteorology and $CO_2$ fluxes from Sodankylä are available at https://zenodo.org/records/12725765. The PhotoSpec observations and $CO_2$ flux observations from CA-Obs are available at https://zenodo.org/records/10048770 and from US-NR1 at https://data.caltech.edu/records/meh5c-wy279. The meteorological data for the North American sites is available from Ameriflux (https://ameriflux.lbl.gov/). The simulation results of SIF are available at https://fmi.b2share.csc.fi/records/8847a0c06c374668b01e345094d373cd.

*Author contributions.*  TT designed the study, implemented Fluspect to version of QUINCY including mSCOPE, performed all the simulations and analysis and was responsible for the first draft of the manuscript. ZP and JS conducted the Photospec observations at CA-Obs, where AB and BJ were responsible for the $CO_2$ flux and meteorological observations. TM and JS were responsible for the PhotoSpec observations at US-NR1. MH conducted the FloX observations at FI-Sod in collaboration with HL. HL contributed the satellite data. MA was responsible for the $CO_2$ flux and meteorological observations at FI-Sod. JPL made original implementation of mSCOPE to the QUINCY
model. TQ provided the L2SM code and help with its use as well as the code for the unit conversion. SZ provided help with the QUINCY model. All the authors contributed to discussing the results and writing of the manuscript.

*Competing interests.*  At least one of the (co)-authors is a member of the editorial board of Biogeosciences..

*Acknowledgements.*  TT acknowledges funding from Research Council of Finland (RESEMON project, grant number 330165; and 337552).
TQ received funding under UKRI NERC grant NE/W006596/1 Structure, Photosynthesis and Light In Canopy Environments (SPLICE) which supported development of the L2SM model. HL and MH acknowledge funding from the Research Council of Finland (grant numbers 337552, 359196, and 353082). We acknowledge the AmeriFlux sites for their data records. In addition, funding for AmeriFlux data resources was provided by the U.S. Department of Energy's Office of Science. We acknowledge the Ministry of Transport and Communications through the Integrated Carbon Observing System (ICOS) research and ICOS Finland. The FloX observations were done as part of European Space
Agency funded project through contract number 4000131497 within the Carbon science Cluster. We thank Tommaso Julitta for help with data processing of the FloX data. Scientific programmers Dr. Jan Engel and Dr. Julia Nabel are thanked for technical support and maintenance of the QUINCY code. TM, ZP and JS acknowledge funding by NASA's Earth Science Division IDS (awards 80NSSC17K0108 at UCLA, 80NSSC17K0110 at JPL) and ABoVE programs (award 80NSSC19M0130). ZP work was supported by a National Science Foundation Graduate Research Fellowship under Grant No. DGE-1650604 and DGE-2034835.



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
