# Peer review of "Modelling sun-induced chlorophyll fluorescence (SIF) in evergreen conifer forests with a terrestrial biosphere model"

_EGUsphere, 2025_

## Referee Comment (RC1)

In this paper, the authors parameterize a sustained component of Non-Photochemichal Quenching for boreal evergreen conifer forests within the QUINCY terrestrial biosphere model (TBM), in the view of improving the representation of leaf-level SIF emission. They implement and evaluate 3 modelling approaches to upscale leaf-level SIF fluxes to the canopy scale. Top-of-canopy (TOC) SIF and GPP simulations are compared at three sites against corresponding *in situ* estimates, as well as, for SIF, coarse spatial resolution space-borne SIF retrievals from TROPOMI at two sites.

Because the utilization of space-borne SIF retrievals for improving the parameterisation of TBMs remains challenging due to the large spatial and temporal scales involved, confronting TBM SIF simulations with *in situ* data appears promising for advancing the realism of process-based representations of how light is partitioned between photosynthesis and energy dissipation (fluorescence and NPQ). Although the study addresses current scientific topics on how SIF data can be used to improve our understanding of the dynamics of plant photosynthesis, I feel that the methodology exhibits shortcomings and inaccuracies which question the significance of the outcomes. While the development of SIF observation operators in other TBMs is typically motivated by the goal of improving their representation of GPP dynamics (through the optimization of specific model parameter values using data assimilation), the limited contextualization provided in this study gives the impression that the implementations around SIF simulation conducted here in QUINCY are more of an academic exercise, disconnected from practical applications while the perspective of data assimilation is raised.

In addition, the structure of the manuscript needs to be improved, in particular regarding the presentation of the different couplings between QUINCY and each radiative transfer (RT) approach. In the current version, the flow of information between the variables simulated by QUINCY and those used by each RT is very unclear; what is common across RTs or what is RT-specific should be more clearly presented. The implementation of a sustained component for NPQ, while relevant for the studied ecosystem is questionable and suffers from an insufficiently rigorous calibration.

At the end, I am left with the impression the take-home messages for the TBM community are difficult to grasp.

Although fit for Biogeosciences, major revisions are required before publication. Below is a list of concerns and comments that support this view.

**Major concerns**

1. The presentation of the coupling between QUINCY and the different modelling approaches is very confusing. The list of variables and parameters of QUINCY that are used by each RT is very not clear. In particular, how are the leaf optical properties (reflectance, transmittance, absorptance) calculated? Is it performed by QUINCY in all cases? How does the leaf nitrogen content translates to leaf chlorophyll content, and how does it impact leaf optical properties, and hence APAR and photosynthetic activity? Do leaf optical properties vary with time and with the canopy depth for all RTs? What is the leaf orientation considered in mSCOPE, and is it consistent with other RTs and with

the one of QUINCY? The spectral dependency of the leaf optical properties and SIF needs to be better explained.

In short, what is common and what is specific to each RT should be presented more clearly. Including a schematic would be beneficial.

Is the 4th modelling approach (§ 2.5.4) really useful? It is acknowledged by the authors as likely unrealistic and the corresponding simulations are largely unexploited in the manuscript. A comparison with the previous implementation (Thum et al., 2017) would have been more relevant.

2. The rationales for the proposed implementation of the sustained component of NPQ for boreal evergreen coniferous forest are questionable. The authors follow the approach of Raczka et al. (2019) for the sustained NPQ model. However, Raczka et al. also developed in parallel a specific parametrisation for the reversible component of NPQ, showing significant impact of how reversible NPQ is represented. The authors use a model that has been calibrated on plant species poorly adapted to the boreal climate that is targeted in this study, and that is then combined to the sustained NPQ model over the whole of the time periods considered. I see a potential risk of partial redundancy in this approach.

The authors could also have considered the reversible model developed in Raczka et al., and assess the impact of their sustained implementation with that of the latter study.

In addition, the presentation of the calibration of the KNs parameters needs significant improvement.

- 3. In relation to the previous sentence, I am somewhat surprised that such an advanced TBM as QUINCY is not equipped with an even simple numerical optimization tool to calibrate some model parameter values in a rigorous manner. Only rough estimates of the values of the newly introduced model parameters are provided, without any indication of their associated error and quantification of the corresponding model-data fit. This is a bit problematic, all the more as incorrect quantification of the leaf-level SIF yield can partly contribute to the overestimation of TOC SIF that is observed whatever the RT modelling approach.
- 4. While the temporal dynamics of the simulations of SIF by QUINCY are consistent with *in situ* data (when the sustained component of NPQ is considered), they however show a large positive bias. This is however not the case for GPP nor APAR which agree fairly well with the *in situ* estimates. Those contrasting results seem to indicate that the SIF parameterisations are incorrect, but it remains unclear whether it is related to the equations implemented or inappropriate parameter values. As written above, the coupling between QUINCY canopy module (which simulates leaf-level SIF emission) and the RT schemes is not detailed enough to allow the reader understanding what is going on. The way canopy architecture is represented in QUINCY (leaf clumping and inclination, in addition to LAI) could partly explain those large biases (as acknowledged by the authors in the Discussion) but their is no clear information of those parameterisations (values of leaf clumping and leaf inclination that could be compared to literature values, etc.).

A comparative sensitivity analysis of SIF and GPP could have provided first order information on the key parameters requiring further calibration (like parameters strongly affecting SIF yet marginally impacting GPP).

The authors conclude that their work "pave the way for data assimilation using SIF observations". This has already been undertaken by other groups to correct GPP. Here, given that GPP is already in good agreement with *in situ* data, what could be the benefit of assimilating SIF observations?

- 5. The paper is a bit long. It contains several redundancies and repetitions that could be avoided. In addition, the chaining of some sections could be re-worked (see comments below) to improve the logical flow with the idea of making the manuscript more concise.
- 6. The referencing needs improvements.

**Title:**

7. The title does not reflect the developments that are specific to this study: It is very similar to a previous paper from the authors in which SIF modelling was implemented in QUINCY for the same ecosystems (Thum et al., 2017).

**Abstract:**

- 8. L2: "...to satellites" > "... to satellite observation footprints"?
- 9. L6: What does "alternative" mean in this context?

**Introduction:**

- 10. L21: Given the focus is on GPP and SIF, information on the GPP budget of these ecosystems would seem more relevant than biomass.
- 11. The structure is a bit odd with the introduction of SIF for monitoring boreal coniferous evergreen forests (L25) before more a general presentation of SIF (L41). Same remark for NPQ, a mechanism that is shared across ecosystems, that is presented after its sustained component, which seems to be specific to boreal evergreen conifer forests. The works of Porcar-Castell et al. (2008 and 2011) on sustained NPQ, prior to the work of Adams et al., should have been cited.
- 12. L37: The sentence about the belowground carbon may be interesting, but it breaks down the chaining between the mention of the impact of climate change and the possible use of TBMs to anticipate the future of ecosystems at high latitudes.
- 13. L44: Several studies prior to the one of Porcar-Castell et al. (2021) have already pointed out and use satellite SIF data to provide information on  $CO_2$  uptake: In addition to the studies cited in the following sentences, the works of Guanter et al. (2012) and Joiner et al. (2011, 2013) using GOSAT and GOME-2 for instance.
- 14. There have been also numerous modelling studies that have highlighted the "complexity" of the SIF-GPP relationship depending on the scales considered. The paper of Parazoo et al. (2020) also highlighted model-data discrepancies at US-NR1.

- 15. L59: "...absorbed", as well as "scattered".
- 16. L60. Canopy structure also explains TOC SIF anisotropy (Middleton et al. 2015; Joiner et al., 2020; Malenovsky et al., 2021 for instance), commonly referred to as "bidirectional" effects. This makes the use of the term "bi-directional modelling" in L75 ambiguous.
- 17. L61: The contribution of the soil to TOC SIF *estimates* is very unclear. How is it accounted for in QUINCY?
- 18. L65: The term "carbon cycle modelling" is ambiguous here, as it is more often associated to global TBMs (those described from L76).
- 19. L65: In Miller et al. (2005), ChlF is simulated by FluorModLeaf.
- 20. L68: A proper reference for SAIL should be one of Verhoef et al.
- 21. L71: "many applications" is too vague.
- 22. The main reason why the TBM community has implemented SIF models is for assimilating satellite SIF data to improve the simulation of GPP. This is really not reflected here, in particular with respect to the cited papers of Koffi et al. or Bacour et al. It is also really unclear if the perspective of assimilation of SIF data is what is motivating the developments made in QUINCY. In addition, I find it a bit strange that QUINCY and its previous SIF model developments are not presented here, associated with the TBM community.

The various satellite data that have been considered in the cited data assimilation studies could be indicated.

Other SIF modelling works (2020)worth mentioning could be: Cui et al. (https://doi.org/10.1016/j.rse.2019.111373) or Wang et al. (2021)al. (https://doi.org/10.1016/j.agrformet.2021.108424) with BEPS, Fan (2025)or et (https://doi.org/10.1029/2024JG008280).

Associating the work of MacBean et al. (2018) to "computational burden" in relation to the modelling of SIF in TBMs is surprising as they considered a simple SIF-GPP relationship.

- 23. L86: "A 2-stream"... the sentence is not clear.
- 24. "... pave the way for data assimilation approaches". 1) If this is the main motivation of this work, it should be introduced earlier, 2) so what about some of the works that are cited earlier? Another aspect of data assimilation, in addition to the definition of an observation operator, is the development of a numerical optimization algorithm capable of tuning specific model parameters to minimize the discrepancy between model simulations and observational data. While I view the model implementations conducted in this study very positively, what I have seen regarding the calibration of the model parameters does not meet the standards expected for such a perspective.

It seems to me that those objectives are very much dependent of QUINCY, all the more they have been already undertaken (partly) with other TBMs. The previous works with QUINCY should be introduced before to highlight what is the novelty of this study.

25. The reference to the TROPOMI data that are used in this study should be provided here, given in particular that two datasets exist.

**Data and model descriptions**

- 26. §2.1. Do the sites belong to the same ecoclimate class? It is not clear if QUINCY uses plant functional types (PFT) classes for simulations, and if so, what is (are) the PFT(s) considered for those three sites. Do they have the same dominant PFT?
- 27. L108: About the red and near-infrared regions, it is unclear if the ranges provided here also correspond to the ones used for QUINCY simulations.
- 28. L112: If there are three vertical scans at CA-Obs, what value is considered in this study?
- 29. L115. A reference paper for the FI-Sod should be provided if possible.
- 30. L120: There are two identical references for Magney et al. (2019) (2019a = 2019b). There is no presentation of a specific SIF spectrum from Hyytiälä data in the paper of Magney et al. In addition, later in the manuscript (in §2.6) the authors refer to Fig1 from Magney et al.: All spectra in this figure exhibit different variation ranges than the one shown in Figure 1 of the current study. Clarifications on the source of the SIF emission spectrum considered here, and its availability, should be provided.
- 31. L123: Which CO2 fluxes? Are they directly "measured" or "estimated"?
- 32. Title of §2.2 should mention "SIF". The paper of Köhler et al. (2018) should be cited before the one of Guanter et al. (2021). The description of the spectral characteristics may be too detailed for the (already long) paper.

As indicated in Guanter et al. (2021), the TROPOSIF retrievals are provided at 740 nm (and not 743). It is unclear whether the authors have used the "instantaneous" retrievals or the "daily averaged" ones. I guess this is the 1st option as they show in Figure 8 daily averages values around noon. This temporal averaging should be indicated here.

What puzzles me, is the spatial scale of 0.5° around the sites that is considered for averaging the satellite data, given the heterogeneity around the flux tower sites. For instance, CA-Obs (analyzed in Figure 8) is surrounded by lakes, with a "big" one about 6 km away from the site. To which extent surfaces different from "boreal evergreen coniferous forests" within the considered satellite footprints can contribute to the averaged values used for model evaluation? Why not considering an averaging at a higher spatial resolution?

Also, why such strict criterion (0.2) on cloud fraction for data selection? A value of 0.8 is used in Guanter et al. (2021). This strongly limits the number of available observations, in particular during winter (which is a specific attention of the study in relation to the impact of sustained NPQ).

33. The structure for the presentation of QUINCY could be improved, in particular by more clearly separating what existed prior to this paper from the developments made in this work. §2.4.1 is included within "§2.4 Leaf level model of chlorophyll fluorescence" but there is no §2.4.2. The reference to Campbell and Norman (1998) for clumping is a bit vague to understand the parameterisations considered here (the paper of Campbell and Norman is about 300 pages long). To make the paper more concise, the features/characteristics of the model for boreal evergreen coniferous forests could be provided here. In particular, for the leaf scattering albedo (instead of §2.7). What is the value of leaf clumping? Is a leaf inclination function considered? If so, what is it?

34. The presentation of the various yields and rate constants follows previous works (van der Tol et al., 2014; Raczka et al., 2019); this part could be shortened, possibly moving bits to the "Supplement" section .

I do not understand the formalism considered here: the unit of a "rate constant" is the inverse of time (s-1). If they are unitless, do the authors refer to "relative" rate constants (see Laisk et al., 1997)? The variable x in eq. 9 is referred to as "degree of light saturation" in the reference papers.

L228: The vegetation species for which the relationship between x and  $K_N$  has been derived in van der Tol et al. (2014) should be clarified (cotton leaves) to highlight one strong assumption of this study (i.e. that a PFT specific parameterisation of "reversible" NPQ is considered, which may not be adapted to boreal conifer ecosystems).

In Table 2, some "variables" are actually "parameters". The statement that they are unitless is incorrect (see for instance Raczka et al. for the case of fluorescence fluxes). The list of parameters and variables is not exhaustive, the value of some parameters is provided in the text (L230 or L250 for instance) while others are missing ( $F_P^0$ ,  $\Phi_P^0$ , etc.), which does not permit a comprehensive understanding of the parameterisation employed. Couldn't Table 2 be expanded, while also distinguishing which parameter values come from the literature and from this study?

The values for the parameters b and T of  $K_{Ns}$  are provided in L246 as well as in L251.

The notation for "sustained" is sometimes "s", some other times "sus". This could be homogeneised.

35. The presentation of the calibration of  $K_{Ns}$  needs improvement.

L242, Raczka et al. also accounted for the same temperature acclimation in their parameterisation of sustained NPQ; however, they complemented this with the parallel development of a specific reversible component. It is not clear to me why this is not considered in this study, all the more this could possibly result in a form of "double counting" of NPQ when  $K_{Ns}$  is not zero.

I do not understand the justification about the use of only the "SIF observations" at US-NR1 for the parameter calibration, also given there is no attempt in this paper to compare the  $K_{Ns}$  obtained in this paper to that of Raczka et al. It is very unclear what are those "SIF observations"; I guess this must relate to some PAM measurements, but they are not presented in the "Data section" for the corresponding sites.

Is the parameter calibration at US-NR1 performed over the whole of the time series or by considering specific time periods?

- 1. L173: The reference "NOA" is a bit odd. Other references to FLUXNET would be more appropriate.
- 36. What is "1/C°" in eq 5?

- 37. Doesn't eq 12 depends on the layer / LAI?
- 38. The coupling of the canopy module of QUINCY with the different RT approaches is very confusing. It is not clear how eq 12 is exploited by each of the RT approach.

It is unclear whether leaf-level SIF emission is considered as a diffuse flux in all cases, and whether mSCOPE represents a directional component of TOC SIF compared to the other RTs: the observation geometry (nadir) is detailed for mSCOPE but not for L2SM and LZ. Do that mean that for L2SM and LZ, the whole upper hemisphere contributes to the simulated TOC SIF? If so, is the approach adapted for a comparison against nadir viewing instruments?

Why such distinction between mSCOPE and the other models? There is a QUINCY-mSCOPE version but no QUINCY-L2SM or QUINCY-LZ, all the more it finally appears that L2SM is the preferred option by the authors.

The spectral dependency of the leaf optical properties and SIF needs to be better explained; it is absent from all the equations. L347: the specific equations should be provided.

What do the "red" and "far red" ranges correspond to? The source of the SIF emission spectrum is unclear (see previous comment) and its application in each case is not detailed enough (what is done for the simulations at 740 nm?).

In eq 21, is SIFinternal somewhat related to SIFcl? If not, how is it defined? What is E in eq 21? Eleaf?

Eq 22 is a very generic equation, which does not seem specific to the LZ approach. Where is the spectral dependency in eq 22, as LZ is used to calculate TOC SIF in the red and far red ranges? Is the "upscaling only" implementation really useful, as it is not realistic and little discussed later in the paper?

- 39. Title of §2.7 could be clarified, as for instance "site scale flux simulation protocol". It is unclear whether some kind of spin-up is performed at the different sites. L358-L360: same remark as before on the way the values for soil albedo and LAI are prescribed. Is the soil albedo the same for the three sites?
- 40. A section dedicated on the description of how the models are evaluated is missing. It could detail typical evaluation metrics considered here (R2, RMSE, etc.). Because it seems that the model bias dominates the error budget, RMSE could also be decomposed into systematic vs random components.

The section could also include how the escape factor is calculated for each RT approach as well as information about the computational efficiency. It appears that the latter criterion is the one the authors ultimately considered for the selection of the optimal RT scheme, but there is no information/number to allow assessing how much "faster" is L2SM compared to mSCOPE. The corresponding analysis should be provided somewhere in the result section.

Finally, this "evaluation methodology" section could also include the description of the specific processing of the data (monthly averaging and calculation of the midday values, etc.), that do not fit in the Result section (L368).

**Results:**

- 41. Because the modelling of sustained NPQ is presented before the RT approaches in the Methods section, I would find it more logical to present the corresponding results on the impact of the accounting of  $K_{NPQs}$  on the simulations first.
- 42. Section 3.1 is way too long. It could be separated into specific subsections (for instance by splitting the evaluation of the escape factor estimates) and redundancies should be avoided. In addition, several discussion of the results would rather fit in a "Discussion" section (ex: L390-L395, L397-L399, L406-L409, etc.) or in the "Methods" section (L406, L420, L433, etc.).
- 43. A figure synthetising the model-data (dis)agreement over all sites for all RT schemes is missing.
- 44. L380: I do not understand the statement about the necessity to have good GPP performance to "successfully" (whatever this means) simulate SIF. One could imagine that, in order to correct for the large bias in SIF simulations, one could simply reduce LAI for instance, which would then degrades the GPP simulation performance.
- 45. L387-L388: The sentence is awkward.
- 46. L406: It is indicated here, in the Result section, that sustained NPQ is not considered for Sodankylä. This should rather be introduced in the "Methods" section. The best would however be to clearly show the impact of the accounting or not of sustained NPQ at the three sites in the first section of the "Results".
- 47. L423: Could the lowest SIF performance at FI-Sod be explained by not accounting for sustained NPQ at this site?
- 48. Table 4: Why considering only the summer values, given one of the focus of the study is more on winter and summer times when the sustained component of NPQ can not be neglected. I would be very much interested in seeing the temporal variation of this variable (which is not common in the literature). Could the estimated values be compared against other works?
- 49. L443: Isn't APAR expressed in μmol of photons?
- 50. L443: I do not understand what "ecosystem response" refers to here.
- 51. L449: "... there was no feedback" > "in the current framework, there is currently no feedback"? This part should be moved to "Discussion".
- 52. L452: I do not understand the causality for the lower seasonal cycle of APAR at US-RN1. Is FAPAR constant for all sites (as it could be interpreted from L329)?
- 53. L457: Is it ΦF or ΦFt?

- 54. §3.3. Why focusing on summer time for this analysis, when the sustained component of NPQ is less important?
- 55. L467: Have the authors tried to fit two separate hyperbolic functions for morning vs afternoon values to support that statement?
- 56. L470: "...lower level". For SIF, GPP or both?
- 57. L473 about the stomatal conductance. Difficult to understand as how it is implemented in QUINCY has not been presented. This part fits better in the Discussion section.
- 58. L476: What does "smaller uncertainties" for the parameter *a* means? I am puzzled that the calibration of the parameters of these hyperbolic functions (used for a rather "qualitative" evaluation) is performed "rigorously" with a dedicated optimization routine, while this is not the case for some of the main parameters of the model.
- 59. L478: I have not understood the sentence.
- 60. L480: Which is the Figure / Table that show the lower performance at US-NR1? Why not providing a Table in which the scores are calculated over the whole of the temporal window for the three sites? Again, why focusing on a limited window in summer?
- 61. L485: Isn't it Table 5 instead?
- 62. L487: About understorey vegetation, this should be moved to the "Discussion" section. Because no information has been provided in the "Methods" section on how the vegetation is described in QUINCY for the different sites (100% boreal evergreen needleleaf forest?), the implications for potential model improvements are difficult to grasp.
- 63. L495-L497: This is more for Discussion.
- 64. §3.4: The title of the section is very misleading and does not reflect what is presented here (a comparison between model simulation and satellite estimates).
- 65. L500: "better agreement"... This should be supported by quantitative metrics.
- 66. L502-L503: A bit awkward given that the averaging characteristics are determined by the authors. I have understood that the authors use daily SIF data; do those "rapid **phenological** changes" occur at a higher temporal resolution? What do "phenological" refer to here? Do the authors mean "physiological"? Note also that the temporal monitoring of SIF by satellite observation is limited by acquisition around noon.
- 67. L503: The authors could have also adopted a less stringent criterion on cloud fraction.

**Discussion:**

- 69. Several pieces of information about modelling assumptions and protocol are provided in the Discussion section. Presenting them earlier in the Methods section would make it easier to interpret the results while reading the manuscript.
- 70. Table S3: The caption is not clear. Couldn't the "overestimation" be calculated as a ratio or as a bias?
- 71. L523: Is this statement supported by other studies?
- 72. L529: Which are those "other ecosystems" and tests, in relation with the model developments presented in this study (sustained NPQ and RT models)?
- 73. L533: Has the assessment of satellite versus *in situ* data not been addressed in previous studies?
- 74. L535: About the relevance of "simple approaches" and the assessment of the modelling SIF performance based only on R2: The conclusion is really dependent on what ones want to do with SIF! If the objective is to assimilate SIF observations to improve TBMs' simulation of GPP, isn't it preferable to use a process-based representation of SIF that is consistent with that of GPP? Given the substantial mismatch between the modelling performance of GPP and SIF in QUINCY, it seems likely that there is an issue with the SIF modelling framework. Under the current model, would it be possible to assimilate SIF data without degrading GPP?
- 75. L544: A more detailed description of the clumping and description of the canopy architecture would be helpful in the section presenting the model.
- 76. L550: I agree with the authors that a consistent representation of the radiation regime within the canopy is required for SIF and GPP. However, as said before, the current way the model is presented does not allow understanding the improvements to be made in that direction. How would the authors "include" NIRv in their framework? Is QUINCY equipped with a RT scheme allowing to simulate the corresponding TOC reflectances as it is the case for SIF? What datasets would they consider? *In situ* or space-borne data? The values of the calculated escape factors could have been compared to literature.
- 77. L564: Could the authors provide references on the variation of SIF emission spectrum across species? Given that all RT approaches yield approximately the same degree of overestimation in SIF simulations compared to observations, how important is this assumption relative to other potential sources of error? It would be helpful if a ranking of these possible sources of error could be provided. What is the relative contribution of incorrect parameter values (which could potentially be corrected through data assimilation) compared to incorrect process representation?

- 78. L567: "... in vivo spectrum". Could a reference be provided? I do not understand the implication of this, compared to the other choices made. Could the authors elaborate? Does this part related to the spectral dependency of SIF really fits within a "radiative transfer" oriented section?
- 79. L569-L585: Same remark as before, this part related to data processing do not fit within this RT section (which is already very long).
- 80. L605: Although leaf chlorophyll content is the main factor of leaf absorptance in the visible, this is not the case in the near-infrared.
- 81. L606-L607: So, this is not the case in QUINCY? What to the authors mean by "include"? Does it relate to the representation of the temporal variations and/or vertical gradients?

**Conclusions:**

- 82. L615: While the study focuses only on QUINCY TBM, this general statement seem to apply to other similar models. Could the authors elaborate on this? What is expected from a better accounting of the canopy structure by proximal sensing measurements?
- 83. L619: I thought that sustained NPQ was not applied at Sodankylä... A clearer presentation of the simulation configurations is needed.
- 84. L620: Do this "process" refer to sustained NPQ? Why "likely"? Isn't air temperature explicitly used already in the formulation of NPQs?
- 85. L620-L622: TROPOSIF is not a satellite, it is a product derived from the observations by the TROPOMI instrument. What does "seems feasible" mean given that this is exactly what is done in this study (Figure 8)?
- 86. L625: On the link between SIF and latent heat, I am really unsure that this fits in the conclusion of this manuscript given this has never been addressed earlier. This could be expended in the discussion.
- 87. L627: Given the concerns I have expressed, I have some reservations about the concluding remark. Does it concern QUINCY or does it apply to other TBMs as well?

**Tables:**

88. Table 1: (°) for the coordinates should be added. LAI and Air temperature vary with time; Are those mean values? The years considered in the study could be indicated.

89. Table 5: Couldn't the presentation of Table 3 be adopted also for this table (R2 and RMSE on the same line)?

**Figures**

- 90. Figure 1: TROPOSIF should be indicated in place of TROPOMI. The wavelength of the SIF product is 740 nm, not 743.
- 91. Figure 2: The values on y-axes on the left should also be in purple, not only the label, to facilitate interpretation. The same unit is repeated multiple times; this could be simplified. The years are missing on the left figures. For improved clarity, it would be more consistent to present the panel labels (a, b, c) prior to their corresponding descriptions.

Metrics quantifying the model-data discrepancy (RMSE, R2) could be provided. These remarks apply to all similar figures.

- 92. Figure 3 Figure 7: Please use "LZ" or "Liu/Zeng" or "Zeng" consistently throughout the manuscript and figures.
- 93. Figure 5. Couldn't the different components of  $K_{NPQ}$  be also presented? "photosynthetically active" is missing. Is it phi\_Ft that is shown here (eq 10)?
- 94. Figure 6: Dots and crosses are not distinguishable.
- 95. Figure 7: "Far-red" should appear more clearly in the corresponding figure insets.
- 96. Figure 8: "TROPOSIF" instead of "TropoSIF". 740 nm should be considered instead of 743 nm. What is the "uncertainty" of those TROPOSIF data averaged at 0.5°? Is this a kind of averaged retrieval error or the standard deviation of all individual retrievals? How does it relates to the "standard deviation" cited in L509?

What does "TROPOSIF values are daily" mean? It is not clear whether the authors have considered the daily corrected variable in the TROPOSIF product or not.

---

## Author Comment (AC1)

**Reply to Reviewer 1**

The reviewer comments are in magenta, the replies to the comment are in black. The changes in the manuscript are in italics.

In this paper, the authors parameterize a sustained component of Non-Photochemichal Quenching for boreal evergreen conifer forests within the QUINCY terrestrial biosphere model (TBM), in the view of improving the representation of leaf-level SIF emission. They implement and evaluate 3 modelling approaches to upscale leaf-level SIF fluxes to the canopy scale. Top-of-canopy (TOC) SIF and GPP simulations are compared at three sites against corresponding in situ estimates, as well as, for SIF, coarse spatial resolution space-borne SIF retrievals from TROPOMI at two sites.

Because the utilization of space-borne SIF retrievals for improving the parameterisation of TBMs remains challenging due to the large spatial and temporal scales involved, confronting TBM SIF simulations with in situ data appears promising for advancing the realism of process-based representations of how light is partitioned between photosynthesis and energy dissipation (fluorescence and NPQ). Although the study addresses current scientific topics on how SIF data can be used to improve our understanding of the dynamics of plant photosynthesis, I feel that the methodology exhibits shortcomings and inaccuracies which question the significance of the outcomes. While the development of SIF observation operators in other TBMs is typically motivated by the goal of improving their representation of GPP dynamics (through the optimization of specific model parameter values using data assimilation), the limited contextualization provided in this study gives the impression that the implementations around SIF simulation conducted here in QUINCY are more of an academic exercise, disconnected from practical applications while the perspective of data assimilation is raised.

One motivation for this study is the possibility to use spaceborn SIF observations for model evaluation at global scale and also to study the carbon and water cycle at different scales, but we had not mentioned this clearly in the earlier version of the manuscript. We will clarify this further in the revised version of the manuscript. QUINCY has been developed with emphasis on the nutrient cycles and once we have SIF implemented inside the model, it will also help in studying the nutrient cycles.

In addition, the structure of the manuscript needs to be improved, in particular regarding the presentation of the different couplings between QUINCY and each radiative transfer (RT) approach. In the current version, the flow of information between the variables simulated by QUINCY and those used by each RT is very unclear; what is common across RTs or what is RT-specific should be more clearly presented. The implementation of a sustained component for NPQ, while relevant for the studied ecosystem is questionable and suffers from an insufficiently rigorous calibration. At the end, I am left with the impression the take-home messages for the TBM community are difficult to grasp.

One of the main motivations for this study was testing of different radiative transfer models to get a view, if a simpler and less computationally demanding approaches would be appropriate. We're sorry that our current manuscript version was unable to convey this message and will try to clarify this in the next version of the manuscript.

Although fit for Biogeosciences, major revisions are required before publication. Below is a list of concerns and comments that support this view.

We thank the reviewer for taking time and effort to go through the manuscript in such a detailed manner and giving insightful comments and suggestions. We're certain that addressing them will greatly improve the quality of the manuscript.

**Major concerns**

1.      The presentation of the coupling between QUINCY and the different modelling approaches is very confusing. The list of variables and parameters of QUINCY that are used by each RT is very not clear.

We will do our best to clarify this in the new version of the manuscript.

In particular, how are the leaf optical properties (reflectance, transmittance, absorptance) calculated?

The leaf optical properties are set per plant functional type in basic QUINCY. The model uses single scattering albedo (SSA, i.e., the sum of reflectance and transmittance) that has one plant functional type (PFT)-specific value for the visible and the near infrared-region. The absorptance is 1 - SSA. We will clarify this in the manuscript.

We had this in the earlier manuscript version as:
"*Radiative transfer is calculated separately for the visible and near-infrared bands. Leaf reflectance is calculated based on the PFT-specific single leaf scattering albedo. Leaf transmissivity is assumed to be equal to reflectivity.*"

We will clarify this by adding a mention of absorptance here. We will also add in here how the soil reflectance was estimated.

Determining the leaf optical properties is different for the mSCOPE calculation, since that model includes PROSPECT (a model for calculating leaf optical spectra).

We had clarified this in the first version of the manuscript as:
"*The vertical profile of leaf chlorophyll, that was calculated inside QUINCY, was used to calculate the radiative properties of each layer.*"

We will extend this by adding a notification of the PROSPECT model.

Is it performed by QUINCY in all cases?

In this respect the mSCOPE model was different, as we needed to include it in QUINCY. The other approaches were calculated outside QUINCY.

How does the leaf nitrogen content translates to leaf chlorophyll content, and how does it impact leaf optical properties, and hence APAR and photosynthetic activity?

This is only relevant for the mSCOPE implementation, which had the optical properties of leaves changing in the canopy. We have added a section explaining the calculation of leaf chlorophyll from the leaf nitrogen to the supplement. In the basic version of QUINCY, the leaf chlorophyll content is not influencing the leaf optical properties. We have now clarified this in the manuscript.

Do leaf optical properties vary with time and with the canopy depth for all RTs?

The optical properties could vary only in the mSCOPE implementation. Because this is a canopy module simulation, the value of leaf nitrogen and therefore of leaf chlorophyll is not changing in time. However, even in the full QUINCY code the seasonal variation of leaf chlorophyll would be small for these ecosystems (e.g. Fig. 3 in Miinalainen et al., 2025, https://doi.org/10.5194/bg-22-6937-2025).

What is the leaf orientation considered in mSCOPE, and is it consistent with other RTs and with the one of QUINCY?

We had written in the manuscript concerning mSCOPE:

"*For the sunlit leaves, the absorbed radiation is calculated for discrete leaf orientations, including 13 leaf inclinations and 36 leaf azimuth angles relative to the solar azimuth.*"

This is the basic set up for mSCOPE and has been described in more detail in the related publication. We will clarify in the text that the other approaches assume hemispheric leaf orientation and are therefore not consistent with mSCOPE in that respect.

The spectral dependency of the leaf optical properties and SIF needs to be better explained.

We will pay attention to these points in the revision of the manuscript.

In short, what is common and what is specific to each RT should be presented more clearly. Including a schematic would be beneficial.

We will pay attention to these remarks in the revised version of the manuscript and will include a schematic.

Is the 4th modelling approach (§ 2.5.4) really useful? It is acknowledged by the authors as likely unrealistic and the corresponding simulations are largely unexploited in the manuscript.

The idea of the "Upscaled" approach was twofold:
1 - To show the that the improvement obtained by the more complicated radiative transfer approaches in terms of r2
2 - To allow approximating the escape fraction and compare between the sites

Since the reviewer did not consider this approach relevant and pointed out that the manuscript is too long, we decided to remove this approach from the new version of the manuscript.

A comparison with the previous implementation (Thum et al., 2017) would have been more relevant.

This point is quite perplexing, as the "previous implementation" mentioned here has been done in a totally different model. It was done in JSBACH, which is a land surface model having a different implementation of the Farquhar model. Also, a different leaf level model for chlorophyll fluorescence was implemented, and as this model was using two branches, a description for sustained non-photochemical quenching was not needed. When the study was conducted, there were no passive site level observations available and therefore the site level evaluation used MONIPAM observations from Hyytiälä and for further evaluation satellite observations were used (for flux sites, but also for the whole Fennoscandinavian region). The radiative transfer was parameterized simply by using SCOPE results, but the right, observable units of SIF were not calculated.

Because that study had a totally different model and the person making the simulations to this study haven't used the site level version of JSBACH since 2017, we don't consider a comparison to that model relevant. We'd also like to point out that the first publication of the QUINCY model was published in 2019.

2.	The rationales for the proposed implementation of the sustained component of NPQ for boreal evergreen coniferous forest are questionable. The authors follow the approach of Raczka et al. (2019) for the sustained NPQ model. However, Raczka et al. also developed in parallel a specific parametrisation for the reversible component of NPQ, showing significant impact of how reversible NPQ is represented. The authors use a model that has been calibrated on plant species poorly adapted to the boreal climate that is targeted in this study, and that is then combined to the sustained NPQ model over the whole of the time periods considered. I see a potential risk of partial redundancy in this approach.
The authors could also have considered the reversible model developed in Raczka et al., and assess the impact of their sustained implementation with that of the latter study.
In addition, the presentation of the calibration of the KNs parameters needs significant improvement.

The way that the SIF is simulated in the Razcka et al. (2019) paper is done uses two methods. The first method (called CLM-NPQ in Fig. 6 of the paper) is the one we have adapted. The other method additionally includes reversible NPQ (called CLM-NPQ-kR in Fig. 6).

The effect of introducing the time-varying reversible NPQ made the results "improve slightly" in the Razcka et al. (2019) paper. Part of this change comes from lowering of the summertime SIF values (which were overestimated by all the simulation approaches of the study), while also the seasonality improved (if interpreting table S2, the r2 changed from 0.87 to 0.93). Larger improvement was seen in terms of RMSE, where it dropped from 0.98 (likely in units W m-2 sr-1 µm-2) to 0.62 (W m-2 sr-1 µm-2), and this is perhaps the "significant impact" that the reviewer is referring to. Judging on Fig. 6 of the Razcka study, this would be caused by the lower summertime level of SIF compared to the larger overestimation by CLM-NPQ. In our case lowering the summertime values would also bring

improvement to our RMSE, but considering the potential uncertainties and the use of discontinuous fitting, we don't see this as a reasonable task.

The fitting of reversible NPQ in the Razcka study was done by using the springtime and autumn periods separately and then transitioning between these fits during the summer time (the evolution is shown in Fig. S4 of the Razcka et al. 2019 paper). This kind of discontinuous parameterization is very challenging to generalize, as for a general model one would want to have continuous parameterizations. Also, the new CLM model implementation for SIF (Li et al., 2022) did not use the Razcka et al. (2019) formulation at their global scale simulations, only in the US-NR1 site simulation. We're aiming for a more generalizable approach than the one using two different temporal fits at one site.

Our model performance at CA-Obs was already at least 0.90 with all the radiative transfer approaches. Trying to add in a fit that was estimated from MONIPAM observations from Hyytiälä (that is changing with time in a way that might be a bit difficult to replicate based on the publication alone) would have been adding another layer of complexity to the model that was not considered necessary. We find it important that the reviewer brought this point to our attention and will add a point about this to the new version of the manuscript.

We haven't claimed that we have calibrated the KNs parameters. We have tuned them by using an ensemble of simulations.

3.      In relation to the previous sentence, I am somewhat surprised that such an advanced TBM as QUINCY is not equipped with an even simple numerical optimization tool to calibrate some model parameter values in a rigorous manner. Only rough estimates of the values of the newly introduced model parameters are provided, without any indication of their associated error and quantification of the corresponding model-data fit. This is a bit problematic, all the more as incorrect quantification of the leaf-level SIF yield can partly contribute to the overestimation of TOC SIF that is observed whatever the RT modelling approach.

We'd like to point out that the first publication of the QUINCY model was in 2019, it is therefore not as mature as the reviewer might have thought. Also, according to the experience of the first author, implementing new parameter values to the numerical optimization method in this type of model (based on experience with the ORCHIDEE model) is not straightforward.

The model parameters of the leaf level model are based on literature. The chlorophyll fluorescence yield that is obtained from the model was in line with other modelling studies (e.g. Razcka et al., 2019, Parazoo et al., 2019). There is therefore no evidence that the lacking calibration of leaf-level parameters of the model leads to model biases that would prevent the interpretation of the results presented here. (See answer to comment #4).

We would like to point out that data assimilation with a model of the complexity of QUINCY (or similar biosphere models) is fraught with issues of equifinality and risks of leakage  of model structural errors onto parameter estimates that severely limit the utility of such calibration exercises targeting only subcomponents of the model. While statistically appealing it is not self-evident that calibrated parameters and their error ranges are

meaningful. We therefore prefer to use an ensemble method, which QUINCY facilitates to evaluate the impact of parameters and select plausible values. In future applications, we might consider the coupling of QUINCY to tools such as LAVENDAR, but for the current study, it is unclear that the application of a DA technique would have yielded more insight.

4.      While the temporal dynamics of the simulations of SIF by QUINCY are consistent with in situ data (when the sustained component of NPQ is considered), they however show a large positive bias. This is however not the case for GPP nor APAR which agree fairly well with the in situ estimates. Those contrasting results seem to indicate that the SIF parameterisations are incorrect, but it remains unclear whether it is related to the equations implemented or inappropriate parameter values.

We'd like to point out that the fluorescence yield from the leaf level model is in line with the other modelling studies (see below) and that would be the only thing controlled by the parameterization of the leaf level ChlF parameters. Therefore, there are two other potential reasons for mismatch between the simulations and the observations: problems with the radiative transfer or the fact that the observation is having so much impact from e.g. soil, that the modelling efforts cannot easily replicate it. The fact that three different approaches for SIF transfer give similar biases would suggest that the latter might be the case. Also the SIF model implementation to CLM5 by Li et al. (2022) shows an overestimation in simulations in the US-NR1 site by the SCOPE and CLM model (Fig. 3) with their default parameterizations.

As written above, the coupling between QUINCY canopy module (which simulates leaf-level SIF emission) and the RT schemes is not detailed enough to allow the reader understanding what is going on. The way canopy architecture is represented in QUINCY (leaf clumping and inclination, in addition to LAI) could partly explain those large biases (as acknowledged by the authors in the Discussion) but their is no clear information of those parameterisations (values of leaf clumping and leaf inclination that could be compared to literature values, etc.).

We have now added the clumping formulation in the manuscript. The leaf inclination in the basic QUINCY is hemispheric,  and this has now been added also to the QUINCY model description. The leaf clumping in basic QUINCY is only taken into account in the radiative transfer of incoming radiation and not yet in e.g. in the L2SM approach. We will make this a point in the discussion that we will look into this in the following steps of the work.

A comparative sensitivity analysis of SIF and GPP could have provided first order information on the key parameters requiring further calibration (like parameters strongly affecting SIF yet marginally impacting GPP).

The only information going from GPP to leaf level ChlF model is the parameters "x", the degree of light saturation. This is determined partly by the amount of nitrogen in the leaf, since that is determining which part of the leaf is light-saturated and which part is not. The fluorescence yield is the variable that is obtained from the leaf level ChlF model. This has been shown in Figs. 5c, S3c and S6c (values between 0.004-0.012). The magnitude of the ChlF yield is similar as was in the Razcka et al. modelling study (Fig. S2c in their study, 0.005-0.013). It is also in line with values from the SIF model comparison study at US-NR1 (Parazoo et al., 2019) and similar to the values shown in the van der Tol et al. (2014) study.

This is not surprising, as all the model parameters that we're having in our ChlF leaf level model are from the literature.

The SIF yield values have been estimated at the US-NR1 from the PhotoSpec observations by dividing the observed SIF value by estimated absorbed PAR (Magney et al., PNAS, 2019). These values were lower, down to 0.001 during winter, up to 0.008 in summer and 0.006 on average during summer. If one would aim for these values for the ChlF yield, the magnitude of the simulated SIF would indeed be smaller. A new study by Pierrat (2024) has measured ChlF yield at US-NR1 and obtained values that were higher than estimates from Magney (up to 0.05). It is therefore not straightforward to say which magnitude the ChlF yield should have at each site. These results anyhow confirm that the overestimation in the simulations was likely caused by the radiative transfer part of the modelling.

Fan et al. (2025) found in their study done with the VISIT-SIF model, that N cycle-related variables were mostly influencing GPP, while it was parameters mostly related to absorbed PAR that were important for SIF. It is likely that the same would be true for QUINCY and we will add this point to the manuscript.

The authors conclude that their work "pave the way for data assimilation using SIF observations". This has already been undertaken by other groups to correct GPP. Here, given that GPP is already in good agreement with in situ data, what could be the benefit of assimilating SIF observations?

QUINCY is a global model and while the development here has been done at site scale, the aim is to use the model at global scale. We do not expect that QUINCY will be as successful at global scale and all ecosystems. An advancement here considering the data assimilation is the finding that a computationally more feasible method for the radiative transfer of SIF calculation works as well as the more computationally expensive one.

Also, the finding that all the sites had a similar bias could in data assimilation be corrected by using a simple scalar factor. However, as data assimilation was only one possible future development for this work,  we have removed the sentence from the end of the manuscript that the reviewer was referencing to, so now the last sentence of this manuscript ends with: "*this work brings QUINCY closer to being a tool for comprehensive analysis of biogeochemical cycles*". We hope that this will now be more in line with the objectives of this current work.

5. The paper is a bit long. It contains several redundancies and repetitions that could be avoided. In addition, the chaining of some sections could be re-worked (see comments below) to improve the logical flow with the idea of making the manuscript more concise.

We will shorten the paper and place some of the material to the supplementary.

6. The referencing needs improvements.

We are grateful to the reviewer for suggesting several references in the comments and we have added all of them to the manuscript and hope that this will improve the referencing.

7. The title does not reflect the developments that are specific to this study: It is very similar to a previous paper from the authors in which SIF modelling was implemented in QUINCY for the same ecosystems (Thum et al., 2017).

As already mentioned earlier, the earlier study from 2017 was not done with QUINCY. But we agree with the reviewer otherwise here and modified the title to reflect this study more to be:

"*Using different radiative transfer schemes for solar-induced chlorophyll fluorescence (SIF) at evergreen coniferous forests*".

**Abstract:**

8. L2: "...to satellites" > "... to satellite observation footprints"?

Thanks, we changed this.

9. L6: What does "alternative" mean in this context?

We replaced "alternative" with "different".

**Introduction:**

10. L21: Given the focus is on GPP and SIF, information on the GPP budget of these ecosystems would seem more relevant than biomass.

We removed the point about biomass.

11. The structure is a bit odd with the introduction of SIF for monitoring boreal coniferous evergreen forests (L25) before more a general presentation of SIF (L41).

We changed the order of the paragraphs so that observations of SIF start the section, followed by modelling and now things related to NPQ_s are in the end, which more reflects the priorities of this study.

Same remark for NPQ, a mechanism that is shared across ecosystems, that is presented after its sustained component, which seems to be specific to boreal evergreen conifer forests.

Re-arranging the paragraphs now changed this so that general NPQ is introduced before sustained NPQ.

The works of Porcar-Castell et al. (2008 and 2011) on sustained NPQ, prior to the work of Adams et al., should have been cited.

We have now added these references.

12. L37: The sentence about the belowground carbon may be interesting, but it breaks down the chaining between the mention of the impact of climate change and the possible use of TBMs to anticipate the future of ecosystems at high latitudes.

We removed that point.

13. L44: Several studies prior to the one of Porcar-Castell et al. (2021) have already pointed out and use satellite SIF data to provide information on CO2 uptake: In addition to the studies cited in the following sentences, the works of Guanter et al. (2012) and Joiner et al. (2011, 2013) using GOSAT and GOME-2 for instance.

We did not refer to Porcar-Castell (2021) directly in connection to satellite SIF, the sentence where this reference in L44 was is: "*SIF is linked to the light reactions of photosynthesis and can therefore provide information on terrestrial CO2 uptake (Porcar-Castell et al., 2021)*".

We had cited Porcar-Castell (2021) because it is a review paper giving an overview of SIF. However, we added the references that the reviewer mentions here to the following sentence, which was about the link between spaceborn SIF observations and photosynthesis.

14. There have been also numerous modelling studies that have highlighted the "complexity" of the SIF-GPP relationship depending on the scales considered. The paper of Parazoo et al. (2020) also highlighted model-data discrepancies at US-NR1.

The reviewer is right, one example of such a study would be Thum et al., (2017)  but it's not clear where this would best fit in. The modelling part is coming later in the manuscript.

We have referenced the Parazoo paper later in the manuscript.

15. L59: "...absorbed", as well as "scattered".

Sure, this has been added.

16. L60. Canopy structure also explains TOC SIF anisotropy (Middleton et al. 2015; Joiner et al., 2020; Malenovsky et al., 2021 for instance), commonly referred to as "bidirectional" effects.

Thanks for this point, we added it to the manuscript.

This makes the use of the term "bi-directional modelling" in L75 ambiguous.

Sure, we replaced this by "two-directional modelling".

17. L61: The contribution of the soil to TOC SIF estimates is very unclear. How is it accounted for in QUINCY?

This depends on the method that is used to calculate the TOC SIF. We will clarify this in the sections where the different radiative transfer approaches are described.

18. L65: The term "carbon cycle modelling" is ambiguous here, as it is more often associated to global TBMs (those described from L76).

That's a good point. We replaced "carbon cycle modelling" with "vegetation modelling."

19. L65: In Miller et al. (2005), ChlF is simulated by FluorModLeaf.

Thanks for that clarification, we've corrected the wrong name.

20. L68: A proper reference for SAIL should be one of Verhoef et al.

The point of the sentence was that SCOPE includes SAIL and only reference to a SCOPE paper was included. We have now added a reference to Verhoef et al. additionally.

21. L71: "many applications" is too vague.

Sure. We modified the original sentence:
"*The SCOPE model has been widely used in many applications (e.g. Damm et al., 2014, Martini et al., 2019, Wang et al., 2023, Zhang et al., 2018).*"
to
"*The SCOPE model has been widely used in many applications, e.g. studying relationship of GPP and SIF in different ecosystems and under different fertilization treatments as well as water stress effects (e.g. Damm et al., 2014, Martini et al., 2019, Wang et al., 2023, Zhang et al., 2018).*"

22. The main reason why the TBM community has implemented SIF models is for assimilating satellite SIF data to improve the simulation of GPP. This is really not reflected here, in particular with respect to the cited papers of Koffi et al. or Bacour et al.
It is also really unclear if the perspective of assimilation of SIF data is what is motivating the developments made in QUINCY.

One could also think that the spaceborn information of SIF could help in model development, as it likely offers one of the best remote sensing based observations related to photosynthesis. With the emergence of extreme events that are challenging to model with TBMs, the remote sensing data could offer information in regions where we don't have ground observations and truly be a valuable tool in model development.

Data assimilation, while very valuable, will in the end give better modelling results with optimized parameters with their uncertainties. It can pinpoint regions and conditions where the model cannot simulate the observations and needs improvement. One could anyhow think that remote sensing observations could be used in model development also without data assimilation (e.g. studies by Miinalainen et al., 2025 and Peano et al., 2025).

Qiu et al (2019, the reference suggested by the reviewer below), stated that :"*The close relationships between GPP and SIF have led to the use of SIF to benchmark GPP*

*representations and other parameters in TBMs and to improve understanding of the coupling between carbon and water cycles (Thum et al., 2017; Qiu et al., 2018).*" Also, the Razcka et al. 2019 state in their abstract: "*An improved mechanistic understanding is necessary to leverage SIF observations to improve representation of ecosystem processes within land surface models.*" So, we believe that there are other modelling groups who also see other value in SIF than only data assimilation.

We did not make this point clear in the original manuscript and will do our best to convey this point in the revised version of the manuscript.

In addition, I find it a bit strange that QUINCY and its previous SIF model developments are not presented here, associated with the TBM community.

As mentioned earlier, this is the first QUINCY related SIF study.

The various satellite data that have been considered in the cited data assimilation studies could be indicated.

We've added to the text:
"*The TBM studies mentioned used spaceborn data from GOSAT (Kuze et al., 2009) and OCO-2 (Frankenberg et al., 2014).*"

Other SIF modelling works worth mentioning could be: Cui et al. (2020) (https://doi.org/10.1016/j.rse.2019.111373) or Wang et al. (2021) (https://doi.org/10.1016/j.agrformet.2021.108424) with BEPS, or Fan et al. (2025) (https://doi.org/10.1029/2024JG008280).

We thank the reviewer of these references and have added them to the manuscript. (The first doi was to a publication by Qiu, so we added that, as we were not sure which publication would be by Cui).

Modified text now says:
"*Qiu et al. (2020) implemented SIF to the BEPS model and emphasized the importance of accounting for scattering in the modelling, and also data assimilation studies have been done with the BEPS model (Wang et al., 2021).*"e

We will add a reference to Fan et al. (2025) later in the manuscript.

Associating the work of MacBean et al. (2018) to "computational burden" in relation to the modelling of SIF in TBMs is surprising as they considered a simple SIF-GPP relationship.

Sure, we removed the reference.

23. L86: "A 2-stream"... the sentence is not clear.

The original sentence was:

*"A two-stream radiative transfer of SIF signal has been considered to be computationally tangible (Sun et al., 2023a) and a recent model (Quaife, 2025) enables calculation of SIF, since it describes radiative transfer of emission originating from the canopy.* "

We apologize for unclear expression and have modified the sentence to:

*"One way to simplify the calculation of SIF signal's radiative transfer would be to use a two-stream radiative transfer model (Sun et al., 2023a). A recent two-stream radiative transfer model (Quaife, 2025) describes radiative transfer of emission originating from the canopy and therefore enables calculation of SIF signal's radiative transfer."*

24. "... pave the way for data assimilation approaches". 1) If this is the main motivation of this work, it should be introduced earlier, 2) so what about some of the works that are cited earlier?

As explained above, this is not the main motivation for the work, but having a more calculationally feasible approach for the radiative transfer of SIF will certainly make data assimilation easier.

We have modified the sentence from:
*"This work aims to improve the modelling of ChlF so that a TBM can fully exploit the potential of the different data streams associated with SIF and pave the way for data assimilation approaches."*
to:
*"This work aims to improve the modelling of ChlF so that a TBM can fully exploit the information content provided by the ChlF related observations at different scales to improve the understanding of ecosystem processes related to biogeochemical cycles."*

Another aspect of data assimilation, in addition to the definition of an observation operator, is the development of a numerical optimization algorithm capable of tuning specific model parameters to minimize the discrepancy between model simulations and observational data. While I view the model implementations conducted in this study very positively, what I have seen regarding the calibration of the model parameters does not meet the standards expected for such a perspective.

Calibration of the model parameters has not been the aim of this study, therefore it is not clear why the reviewer mentions parameter calibration. We have taken most of the parameters from literature or used the default parameters in the models (e.g. in case of mSCOPE). The only parameters that we tuned were related to the sustained NPQ and (by experience from previous data assimilation activities,) such non-linear dynamics are challenging to address adequately with traditional data assimilation methods. QUINCY's parameter interface allows for systematic perturbation of parameters and therefore coupling to data assimilation tools such as LAVENDAR, however, such application would be a topic for another paper.

It seems to me that those objectives are very much dependent of QUINCY, all the more they have been already undertaken (partly) with other TBMs.

It is great to see all the improvements in this field during the last years. Some of the objectives have indeed been addressed by other modelling groups. As take-home messages for the modelling community working also with other models we'd hope there to be that the L2SM is a relatively simple approach to tackle the radiative transfer issue. Also we are having for the first time these three coniferous sites in the same study.

The previous works with QUINCY should be introduced before to highlight what is the novelty of this study.

As mentioned several times earlier, there has not been previous works with QUINCY related to SIF.

25. The reference to the TROPOMI data that are used in this study should be provided here, given in particular that two datasets exist.

Sure, we've added this.

**Data and model descriptions**

26. §2.1. Do the sites belong to the same ecoclimate class? It is not clear if QUINCY uses plant functional types (PFT) classes for simulations, and if so, what is (are) the PFT(s) considered for those three sites. Do they have the same dominant PFT?

We had written in the manuscript "*The Canadian and Finnish sites are in the boreal zone, and Niwot Ridge is a subalpine forest.*" We're not sure what other ecoclimate class information the reviewer would like to have.

We added the fact that QUINCY uses PFT classification to the model description part and that all of the three sites have the same PFT ("boreal needleleaf evergreen forest") in the modelling protocol part.

27. L108: About the red and near-infrared regions, it is unclear if the ranges provided here also correspond to the ones used for QUINCY simulations.

QUINCY has only two bands, one for visible (300-700 nm) and near-infrared band (700 nm-3000 nm). We will clarify this in the model description of QUINCY. Different from QUINCY, the mSCOPE-QUINCY is calculating each wavelength separately, as it includes PROSPECT.

28. L112: If there are three vertical scans at CA-Obs, what value is considered in this study?

Here we have taken the average of observations. We have now clarified this in the manuscript.

29. L115. A reference paper for the FI-Sod should be provided if possible.

We had two reference papers (Thum et al. (2007) and Knorr et al. (2025)) in the beginning of the section, similarly placed as the references for the North American sites.

30. L120: There are two identical references for Magney et al. (2019) (2019a = 2019b).

We apologize for the mistake with the references. We have now corrected these references.

There is no presentation of a specific SIF spectrum from Hyytiälä data in the paper of Magney et al.

Apologies for unclarity in this topic. The dataset for the SIF spectrum was in the dataset connected to that paper (as mentioned in the Acknowledgements of that paper), and this dataset has its own doi. We had made a mistake in not directly referring to the doi of the dataset. This has now been corrected in the manuscript.

In addition, later in the manuscript (in §2.6) the authors refer to Fig1 from Magney et al.: All spectra in this figure exhibit different variation ranges than the one shown in Figure 1 of the current study.

Apologies for the unclear reference. We meant to refer to Fig. 1 in this manuscript, not the Magney et al paper figure. We have now clarified this in the text.

Clarifications on the source of the SIF emission spectrum considered here, and its availability, should be provided.

We hope that the corrections described above will solve this.

31. L123: Which CO2 fluxes? Are they directly "measured" or "estimated"?

We changed this to "*Net ecosystem exchange of CO2 was measured by the eddy covariance method.*"

32. Title of §2.2 should mention "SIF".

We've added SIF to the title.

The paper of Köhler et al. (2018) should be cited before the one of Guanter et al. (2021).

Ok, we changed the order.

The description of the spectral characteristics may be too detailed for the (already long) paper.

We removed the sentence of the spectral characteristics.

As indicated in Guanter et al. (2021), the TROPOSIF retrievals are provided at 740 nm (and not 743).

As mentioned in the manuscript, we were using Level 2 product, which has wavelength of 743 nm (page 11,

https://data-portal.s5p-pal.com/product-docs/troposif/S5P_PUM_PAL_NOV_UPV_SIF_v10.pdf).

It is unclear whether the authors have used the "instantaneous" retrievals or the "daily averaged" ones. I guess this is the 1st option as they show in Figure 8 daily averages values around noon. This temporal averaging should be indicated here.

We apologize for the unclarity. We have used instantaneous retrievals and taken daily averages. We have now added this information to this section.

What puzzles me, is the spatial scale of 0.5° around the sites that is considered for averaging the satellite data, given the heterogeneity around the flux tower sites. For instance, CA-Obs (analyzed in Figure 8) is surrounded by lakes, with a "big" one about 6 km away from the site. To which extent surfaces different from "boreal evergreen coniferous forests" within the considered satellite footprints can contribute to the averaged values used for model evaluation? Why not considering an averaging at a higher spatial resolution?

As mentioned in the manuscript the pixel for the TROPOMI instrument is 3.5 x 5.5 km². It is therefore very difficult to remove the influence of these lakes, as they can likely be inside the pixel. One can perhaps assume that most of the variation seen in the seasonal cycle of TROPOMI-observed SIF stems from the vegetation.

Averaging at 0.5° around the sites allows having enough observations so that the  influence of random errors is not large and also ensuring that there will be enough observations. To address the concern by the reviewer, we calculated how much of evergreen needleaf forests were included in the TROPOSIF observations (based on MODIS land cover classification). This was 26 % for the CA-Obs site. 65 % of the region was classified as "woody savannah", which has 30-60% of tree cover. We will add this in the text and also the corresponding numbers for the FI-Sod site.

One way to estimate the SIF signal originating from the site is to check if the measurement site is inside the TROPOMI pixel. We made a plot for this (Fig. R1 at the end of this document), which shows clearly that having a tight co-location requirement for the observations makes interpretation of the signal challenging. (The land cover for this site was "woody savannah").

Also, why such strict criterion (0.2) on cloud fraction for data selection? A value of 0.8 is used in Guanter et al. (2021). This strongly limits the number of available observations, in particular during winter (which is a specific attention of the study in relation to the impact of sustained NPQ).

This is a valid point that the reviewer is raising. In the presence of clouds the interpretation of the measured SIF signal is more difficult. Also, the amount of stray light is much larger in cloudy conditions and further influences the measurement. The 0.2 threshold for cloud fraction has been recommended, if one doesn't want to use cloud fraction data in the analysis additionally and wants to have more accurate value (quote from Guanter et al., ESSD 2021: "A *strict cloud filtering (e.g., cloud fraction lower than 0.2) should be applied if an accurate SIF mean value is needed, whereas a more relaxed cloud filter (e.g., cloud*

*fraction lower than 0.8) could be applied if smooth spatial and/or temporal signals are needed. The second case is often preferred for SIF, but it must be taken into account that unfiltered clouds can also introduce noise-like changes in SIF time series. Including cloud fraction data in the analysis would be needed for a proper interpretation of the SIF signals if a relaxed cloud filtering was applied")*.

We will now mention this in the new version of the manuscript.

33. The structure for the presentation of QUINCY could be improved, in particular by more clearly separating what existed prior to this paper from the developments made in this work.

We hope that moving the leaf level model to the supplement (that was a novel development for this study, but includes nothing special to QUINCY, as it has been taken directly from literature) will help to clarify this.

§2.4.1 is included within "§2.4 Leaf level model of chlorophyll fluorescence" but there is no §2.4.2.

Yes, that's right, we had used the subsection 2.4.1 to denote the sustained non-photochemical quenching. Now the section has been re-arranged.

The reference to Campbell and Norman (1998) for clumping is a bit vague to understand the parameterisations considered here (the paper of Campbell and Norman is about 300 pages long).

We had mentioned that more detail on the QUINCY model can be found in Thum et al. (2019), where this equation is represented. We have now added this equation to the model description of QUINCY.

To make the paper more concise, the features/characteristics of the model for boreal evergreen coniferous forests could be provided here. In particular, for the leaf scattering albedo (instead of §2.7).

We apologize for the unclear presentation. We will make this.

What is the value of leaf clumping?

We have added the equation to the model description section.

Is a leaf inclination function considered? If so, what is it?

No, the leaf distribution is assumed to be hemispherical in the original radiative transfer code of QUINCY.

34. The presentation of the various yields and rate constants follows previous works (van der Tol et al., 2014; Raczka et al., 2019); this part could be shortened, possibly moving bits to the "Supplement" section .

We agree with the reviewer and have moved this section to supplement.

I do not understand the formalism considered here: the unit of a "rate constant" is the inverse of time (s-1). If they are unitless, do the authors refer to "relative" rate constants (see Laisk et al., 1997)?

Indeed, these are relative rate constants. We have changed in the table them to "relative".

The variable x in eq. 9 is referred to as "degree of light saturation" in the reference papers.

Thanks, we have added that.

L228: The vegetation species for which the relationship between x and KN has been derived in van der Tol et al. (2014) should be clarified (cotton leaves) to highlight one strong assumption of this study (i.e. that a PFT specific parameterisation of "reversible" NPQ is considered, which may not be adapted to boreal conifer ecosystems).

Sure, we have now added this point.

In Table 2, some "variables" are actually "parameters".

Thanks for noticing this, we changed the caption.

The statement that they are unitless is incorrect (see for instance Raczka et al. for the case of fluorescence fluxes).

The reviewer is right that these are not exactly unitless. These are most often referred to as having relative units, "r. u.", in the literature. The flux unit presented by Razcka is not easy to find elsewhere in literature, so we will use the most common way to describe this.

The list of parameters and variables is not exhaustive, the value of some parameters is provided in the text (L230 or L250 for instance) while others are missing (F0P , Φ0P , etc.), which does not permit a comprehensive understanding of the parameterisation employed.

The parameters included in this table are from the literature. Everything that we used in the leaf level ChlF introduced in this is from literature. However, we will add the mentioned parameters and variables to make the list comprehensive.

Couldn t Table 2 be expanded, while also distinguishing which parameter values come from the literature and from this study?

All the parameters come from literature. We have now added this information to the caption of Table 2.

The values for the parameters b and T of KNs are provided in L246 as well as in L251.

Thanks for noticing this repetition. We removed the later occurrence.

The notation for "sustained" is sometimes "s", some other times "sus". This could be homogeneised.

Thank you, this is a very good remark. We have now homogenized this.

35. The presentation of the calibration of KNs needs improvement. L242, Raczka et al. also accounted for the same temperature acclimation in their parameterisation of sustained NPQ; however, they complemented this with the parallel development of a specific reversible component. It is not clear to me why this is not considered in this study, all the more this could possibly result in a form of "double counting" of NPQ when KNs is not zero.

We have answered to this concern above (comment #2).

I do not understand the justification about the use of only the "SIF observations" at US-NR1 for the parameter calibration, also given there is no attempt in this paper to compare the KNs obtained in this paper to that of Raczka et al. It is very unclear what are those "SIF observations"; I guess this must relate to some PAM measurements, but they are not presented in the "Data section" for the corresponding sites.

We took a formulation from literature for this process and only used the SIF observations at US-NR1 to tune the parameter values. The values are not the same as in the Razcka et al. paper, as this is a different model.

Is the parameter calibration at US-NR1 performed over the whole of the time series or by considering specific time periods?

We haven't claimed that we'd have done parameter calibration. We tuned the three parameters (we had not mentioned tuning of Kns,max in the earlier version) considering the whole time series and have now added this information to the manuscript.

1. L173: The reference "NOA" is a bit odd. Other references to FLUXNET would be more appropriate.

The reviewer is right. We changed this reference to Baldocchi et al. (2001).

36. What is "1/C°" in eq 5?

That is the unit for the parameter 0.0301. We corrected that to: "1/°C".

37. Doesn t eq 12 depends on the layer / LAI?

For the case that was "upscaled" in the first version of the manuscript, yes. But this is now a bit too simplified presentation for the mSCOPE and the L2SM cases, as they both include attenuation inside the leaf. We have now changed the equation so that this would now be the internal SIF emission. We have now clarified this in the text.

38. The coupling of the canopy module of QUINCY with the different RT approaches is very confusing. It is not clear how eq 12 is exploited by each of the RT approach.

We have now tried to clarify this topic in the new version of the manuscript.

It is unclear whether leaf-level SIF emission is considered as a diffuse flux in all cases, and whether mSCOPE represents a directional component of TOC SIF compared to the other RTs: the observation geometry (nadir) is detailed for mSCOPE but not for L2SM and LZ. Do that mean that for L2SM and LZ, the whole upper hemisphere contributes to the simulated TOC SIF? If so, is the approach adapted for a comparison against nadir viewing instruments?

Leaf-level SIF emission is considered as diffuse flux in all cases, we will now clarify this in the manuscript. The emission for L2SM and LZ will be isotropic to the whole upper hemisphere, but division by pi ensures that it is calculated only for one direction. L2SM and LZ aren't designed to solve the angular scattering problem.

Why such distinction between mSCOPE and the other models? There is a QUINCY-mSCOPE version but no QUINCY-L2SM or QUINCY-LZ, all the more it finally appears that L2SM is the preferred option by the authors.

mSCOPE needed to be included in QUINCY, the other approaches were calculated outside QUINCY. This was necessary because mSCOPE uses a hyperspectral approach. We will clarify this better in the new version of the manuscript.

The spectral dependency of the leaf optical properties and SIF needs to be better explained; it is absent from all the equations.

L2SM does calculations separately for the visible and the near-infrared regions. We will clarify this in the new version of the manuscript. The LZ approach is also calculated for different regions separately and we had tried to convey this by adding the "reg" subscript to the equation describing this, but as this was not clear enough, we will further work on clarifying this.

L347: the specific equations should be provided.

Sure, they are now included.

What do the "red" and "far red" ranges correspond to?

This approach was done for the corresponding wavelengths, which was dependent on the observation (i.e. it was different for PhotoSpec and FloX).

The source of the SIF emission spectrum is unclear (see previous comment) and its application in each case is not detailed enough (what is done for the simulations at 740 nm?).

We added more explanation on how the spectrum was applied in section 2.6 and hope that this clarifies the issue. We hope this also makes clear how we applied the approach for the 743 nm.

In eq 21, is SIFinternal somewhat related to SIFcl? If not, how is it defined?

Thanks for paying attention to this, as this was now quite illogical. Indeed, the SIF_cl introduced earlier should have been SIF_internal. We will make this change in the manuscript.

What is E in eq 21? Eleaf?

Yes, exactly, thanks for noticing this. We've corrected the equation.

Eq 22 is a very generic equation, which does not seem specific to the LZ approach. Where is the spectral dependency in eq 22, as LZ is used to calculate TOC SIF in the red and far red ranges?

Eq. 22 is calculated separately for the visible and near-infrared regions in the model. We will now mention this in the text and will move the equation to the beginning of this section.

The red and far-red ranges are estimated by using the different values for leaf reflectance for the LZ approach, so only the escape fraction is calculated separately for the different wavelength regions.

Is the "upscaling only" implementation really useful, as it is not realistic and little discussed later in the paper?

We have removed this approach.

39. Title of §2.7 could be clarified, as for instance "site scale flux simulation protocol".

Sure, we used the suggestion provided by the reviewer.

It is unclear whether some kind of spin-up is performed at the different sites.

The simulations were done with the canopy module of the QUINCY model, that involves no spinup (otherwise e.g. LAI and leaf nitrogen content would be calculated in the model and they could not be set-up as was done here). The fact that there was no spinup was missing from this section. We apologize for that and add that information in.

L358-L360: same remark as before on the way the values for soil albedo and LAI are prescribed. Is the soil albedo the same for the three sites?

The default soil albedo in QUINCY is the same for all sites, we've added this to the manuscript.

40. A section dedicated on the description of how the models are evaluated is missing. It could detail typical evaluation metrics considered here (R2, RMSE, etc.).

Sure, we will add this section.

Because it seems that the model bias dominates the error budget, RMSE could also be decomposed into systematic vs random components.

This is a very good idea. We will not use this everywhere, but will use this in a table that combines the results from the different sites together (requested in comment #43).

The section could also include how the escape factor is calculated for each RT approach as well as information about the computational efficiency. It appears that the latter criterion is the one the authors ultimately considered for the selection of the optimal RT scheme, but there is no information/number to allow assessing how much "faster" is L2SM compared to mSCOPE. The corresponding analysis should be provided somewhere in the result section.

As written already above, we decided to remove the escape factor calculation from the manuscript. Sure, we'll add computational efficiency information in the Methods and Results sections.

Finally, this "evaluation methodology" section could also include the description of the specific processing of the data (monthly averaging and calculation of the midday values, etc.), that do not fit in the Result section (L368).

Sure, we will add these points to the new evaluation methodology section.

**Results:**

41. Because the modelling of sustained NPQ is presented before the RT approaches in the Methods section, I would find it more logical to present the corresponding results on the impact of the accounting of KNPQs on the simulations first.

The reviewer has a good point. However, it was the different radiative transfer schemes of SIF that were the most important topic of this topic, and this is why we'd like to keep that as the first topic of the Results section. To overcome the issue raised by the reviewer, we decided to move the modelling description from the Methods section to the supplementary material (as we had done nothing new there, but were only referring to literature) and introduce the sustained NPQ only after introducing the different radiative transfer methods. We hope that this arrangement is making sense.

42. Section 3.1 is way too long. It could be separated into specific subsections (for instance by splitting the evaluation of the escape factor estimates) and redundancies should be avoided.

We removed the section including the escape factor estimates. We have added subtitles for the different sites and tried to shorten the text in this section.

In addition, several discussion of the results would rather fit in a "Discussion" section (ex: L390-L395, L397-L399, L406-L409, etc.) or in the "Methods" section (L406, L420, L433, etc.).

We thank the reviewer for this advice and will do the recommended changes.

43. A figure synthetising the model-data (dis)agreement over all sites for all RT schemes is missing.

We will make a table, as that is often an easier way to convey this kind of information and will add in systematic and random RMSE values, as suggested by the reviewer in comment #40.

44. L380: I do not understand the statement about the necessity to have good GPP performance to "successfully" (whatever this means) simulate SIF. One could imagine that, in order to correct for the large bias in SIF simulations, one could simply reduce LAI for instance, which would then degrades the GPP simulation performance.

The point we tried to convey with that formulation is that if our GPP model was wrong, it would be senseless to try to get meaningful results from the SIF model. But the reviewer is right, we could degrade the performance of the GPP model by reducing LAI. We removed this sentence.

45. L387-L388: The sentence is awkward.

The sentence that the reviewer is referring to, is:
"*Overall, the modelling approaches for SIF were more successful in the far-red region than in the red region.*"

We apologize for the awkwardness and have modified the sentence to be:
"*Overall, modelling of SIF was more successful in the far-red region than in the red region.*"

46. L406: It is indicated here, in the Result section, that sustained NPQ is not considered for Sodankylä. This should rather be introduced in the "Methods" section.

Sure, we added the point in there (Section "Site scale flux simulation protocol").

The best would however be to clearly show the impact of the accounting or not of sustained NPQ at the three sites in the first section of the "Results".

The sustained NPQ was not the main topic of this study, therefore we want to concentrate on the most relevant issue in the beginning of the Results, which was the comparison of the different radiative transfer codes of SIF.

47. L423: Could the lowest SIF performance at FI-Sod be explained by not accounting for sustained NPQ at this site?

We wrote in that sentence that modelling of SIF was less successful than GPP at FI-Sod. Modelling of SIF was anyhow more successful than at US-NR1, and comparable to the performance obtained at CA-Obs. Also the GPP had the best performance metrics at this site. The SIF observations at FI-Sod do not cover the springtime period that would be relevant for sustained NPQ.

The model performance at FI-Sod in relation to the other sites will be made clearer with the new table suggested by the reviewer that combines the results from all the sites.

48. Table 4: Why considering only the summer values, given one of the focus of the study is more on winter and summer times when the sustained component of NPQ can not be neglected. I would be very much interested in seeing the temporal variation of this variable (which is not common in the literature). Could the estimated values be compared against other works?

On the suggestion provided by the reviewer earlier, we decided to remove this table, as the simulation needed to make this table was not considered important.

49. L443: Isn't APAR expressed in μmol of *photons*?

Well, one can also express that in those units, so we added here "photons" as suggested by the reviewer, even though that is not as much used in the literature.

50.L443: I do not understand what "ecosystem response" refers to here.

The reviewer is referring to sentence:
"*The increase towards summer values in aPAR started earlier than the ecosystem response, as the low temperatures prevented the spring recovery of vegetation.*"

We apologize for unclear wording and have modified this to:
"*The increase towards summer values in aPAR started earlier than for the observed GPP, as the low temperatures prevented the spring recovery of vegetation.*"

51. L449: "... there was no feedback" > "in the current framework, there is currently no feedback"?

Thanks, we modified this according to the suggestion.

This part should be moved to "Discussion".

We have moved it to Discussion.

52. L452: I do not understand the causality for the lower seasonal cycle of APAR at US-RN1.

The sentence in the manuscript and the "causality" was:
"*As US-NR1 is more southerly than CA-Obs, the absorbed PAR did not show a pronounced seasonal cycle (Fig. S3a). This was reflected in the simulated SIF, so that the simulation without NPQs showed no clear seasonal cycle. Using the formulation for NPQs gave the simulated SIF a seasonal cycle,...*"

We apologize for unclear expression and have now modified this to:
"*As US-NR1 is more southerly than CA-Obs, the absorbed PAR did not show a pronounced seasonal cycle (Fig. S3a). The simulated SIF without NPQs followed the seasonal cycle of*

*absorbed PAR. In order to simulate the seasonal variation seen in the SIF observations, it was necessary to include NPQs in the modelling."*

Is FAPAR constant for all sites (as it could be interpreted from L329)?

L329 says:
*"To estimate the escape fraction from QUINCY in this way, we used the modelled vegetation reflectance, fAPAR and the single-leaf scattering albedo constants."*

We modified this to:
*"To estimate the escape fraction from QUINCY in this way, we used the modelled vegetation reflectance, fAPAR and the constants for single-leaf scattering albedo."*

53. L457: Is it ΦF or ΦFt?

Indeed, it should be ΦFt. We have now corrected this in the text.

54.§3.3. Why focusing on summer time for this analysis, when the sustained component of NPQ is less important?

Sustained NPQ was not the main focus of our study. Our aim here was to bring together all the sites in one figure to see if this relationship differs between the sites in the observations or the simulations. The SIF-GPP is expected to vary between seasons (e.g. Thum et al., 2017; Yang et al., 2022), and the different sites of the study are located in different regions and would therefore require separate plots.

55. L467: Have the authors tried to fit two separate hyperbolic functions for morning vs afternoon values to support that statement?

As the manuscript was too lengthy, we decided to move this perspective from the revised version of the manuscript and not show the morning and afternoon values separately.

56. L470: "...lower level". For SIF, GPP or both?

Since we're removing the separation between morning and afternoon values in the figure, we will remove this sentence.

57. L473 about the stomatal conductance. Difficult to understand as how it is implemented in QUINCY has not been presented. This part fits better in the Discussion section.

We have added a description of the stomatal conductance in QUINCY in the model description part of the manuscript. The stomatal conductance influences photosynthesis directly in QUINCY, while there is no direct link between stomatal conductance and SIF. Difficulty of capturing drought responses in simulation of SIF has been noticed also in other studies (Miyachi et al., 2025).

58. L476: What does "smaller uncertainties" for the parameter a means?

The reviewer is referring to text where it said:
"*The fitted lines are shown in Fig. 6 and the fitted parameter values with their associated uncertainties are shown in Table S4. The simulated values had generally smaller uncertainties for the fitted parameter a, …*"

We were referencing with "*smaller uncertainties for fitted parameter a*" to the values shown in Table S4. As this seemed to be a qualitative and unclear way to express goodness of fit, we calculated $r^2$ and RMSE to assess the goodness of fit. Therefore we modified this sentence (see below, comment #59).

I am puzzled that the calibration of the parameters of these hyperbolic functions (used for a rather "qualitative" evaluation) is performed "rigorously" with a dedicated optimization routine, while this is not the case for some of the main parameters of the model.

As the reviewer here is pointing out, it is much easier to make a fit to a function outside a complicated model that includes thousands lines of code and is run on HPC.

59. L478: I have not understood the sentence.

The sentence was:
"*The simulated values had generally smaller uncertainties for the fitted parameter a (Table S4), demonstrating the curvilinear relationship between the variables in the simulations that was not as pronounced in the observations, which can be partly be contributed to the high variability in the observations.*"

We apologize for unclear and qualitative expressions. We have modified this into:

"*Goodness of the hyperbolic fits (Table S4) were better for the simulated GPP vs. SIF relationship than for the observed (averaged over three sites $r^2$= 0.73 for the simulated, $r^2$=0.45  for the observed.) Also the RMSE of the fit was smaller for the simulations vs. observations for all the three sites. The worst behaviour of fits (in terms of $r^2$) happened at FI-Sod for the observations, potentially reflecting the fact that the GPP vs. SIF relationship is quite linear for that site.*"

60. L480: Which is the Figure / Table that show the lower performance at US-NR1? Why not providing a Table in which the scores are calculated over the whole of the temporal window for the three sites? Again, why focusing on a limited window in summer?

The information could have been seen in Fig. S2 vs. Figs. 4 and S5 or Table S1 vs. Tables 3 and S2. The figures showing the different RT approaches had metrics calculated over the whole time period. As mentioned before, we will add a table that shows $r^2$ and RMSE for all the sites.

61. L485: Isn t it Table 5 instead?

Yes, thanks for noticing this. This has now been corrected.

62. L487: About understorey vegetation, this should be moved to the "Discussion" section.

Sure, we have now moved this point (including also canopy shading and turbulence conditions) to the "Discussion".

Because no information has been provided in the "Methods" section on how the vegetation is described in QUINCY for the different sites (100% boreal evergreen needleleaf forest?), the implications for potential model improvements are difficult to grasp.

We apologize for the incomplete Methods section in the earlier version of the manuscript and have now revised that section.

63. L495-L497: This is more for Discussion.

Sure, we moved that part to Discussion.

64. §3.4: The title of the section is very misleading and does not reflect what is presented here (a comparison between model simulation and satellite estimates).

We apologize for the misleading title. We have now changed title to:
"*Comparison of simulated SIF to satellite observations*"

65. L500: "better agreement"... This should be supported by quantitative metrics.

We had written some of this below but will change the order and clarify the sentence.

66. L502-L503: A bit awkward given that the averaging characteristics are determined by the authors.

The sentence that the reviewer is referring to here is:
" *In 2019, the seasonal cycle of TROPOSIF was smoother than that of the simulated SIF and PhotoSpec observations at the site, reflecting the fact that satellites have a limited capacity to detect rapid phenological changes given the spatial and temporal averaging.* "

We had added a 15-day averaging to the site level observations to make the seasonal cycle clearer by smoothing the day-to-day variation. We didn't want to do the same averaging to the satellite data to show its variation more clearly.

I have understood that the authors use daily SIF data; do those "rapid phenological changes" occur at a higher temporal resolution? What do "phenological" refer to here? Do the authors mean "physiological"?

Phenology in this context refers to the changes that occur in the vegetation during the shoulder seasons. The temporal resolution used with phenological changes are from days to weeks, usually. So the "rapid phenological changes" would not occur at higher temporal resolution than days. We will do our best to clarify this point in the manuscript.

Note also that the temporal monitoring of SIF by satellite observation is limited by acquisition around noon.

That's a good point and because of this we had plotted here only the midday values from the model results, as had been written in the caption.

67. L503: The authors could have also adopted a less stringent criterion on cloud fraction.

We have answered to this point above (comment #32).

68. L510-L514: Why not showing the corresponding time series?

Sure, we'll add a figure of the FI-Sod observations with the TROPOSIF product to the supplementary material.

**Discussion:**

69. Several pieces of information about modelling assumptions and protocol are provided in the Discussion section. Presenting them earlier in the Methods section would make it easier to interpret the results while reading the manuscript.

We apologize for the unclarity in the structure and will revise the manuscript.

70. Table S3: The caption is not clear. Couldn t the "overestimation" be calculated as a ratio or as a bias?

We have clarified the caption. Sure, the overestimation could also be calculated in another way, we decided to use this way.

71. L523: Is this statement supported by other studies?

The reviewer is referring to the statement:
"*It may therefore be that the characteristics of the in situ sampling in this type of ecosystem provide smaller SIF signals than expected.*"

We had tried to support this sentence in the earlier sentences, which described how earlier modelling studies have also overestimated site level observations.

72. L529: Which are those "other ecosystems" and tests, in relation with the model developments presented in this study (sustained NPQ and RT models)?

We have tested the L2SM approach against grassland and broadleaf trees. This was not connected to sustained NPQ, but to see whether the overestimation of the simulated SIF would be as pronounced in those other ecosystems. We will revise this in the revised version.

73. L533: Has the assessment of satellite versus in situ data not been addressed in previous studies?

For the CA-Obs site, such comparison has been shown by Cheng et al. (2022), where the daily averaged SIF from PhotoSpec was compared to SIF_dc product from Köhler et al. (2018) (Figure C.7). The TROPOMI product was showing up to 20% higher values than the PhotoSpec.

For our knowledge, no such study has been done for the FI-Sod site.

74. L535: About the relevance of "simple approaches" and the assessment of the modelling SIF performance based only on R2: The conclusion is really dependent on what ones want to do with SIF!

We totally agree with the reviewer.

If the objective is to assimilate SIF observations to improve TBMs simulation of GPP, isn t it preferable to use a process-based representation of SIF that is consistent with that of GPP? Given the substantial mismatch between the modelling performance of GPP and SIF in QUINCY, it seems likely that there is an issue with the SIF modelling framework. Under the current model, would it be possible to assimilate SIF data without degrading GPP?

The mismatch is consistent across the sites. If one would assimilate SIF data, there would likely be a scalar estimated that would be used (this is the approach taken by the D&B model). Since the overestimation is similar across sites, one could fit one scalar to tune down the coniferous forest PFT values.

75. L544: A more detailed description of the clumping and description of the canopy architecture would be helpful in the section presenting the model.

We have these in the revised version of the manuscript and hope that they will be helpful in this respect.

76. L550: I agree with the authors that a consistent representation of the radiation regime within the canopy is required for SIF and GPP. However, as said before, the current way the model is presented does not allow understanding the improvements to be made in that direction.

We hope that the revised manuscript will be clearer in this aspect and thank the reviewer for all the helpful and constructive comments that have allowed us to improve the manuscript.

How would the authors "include" NIRv in their framework? Is QUINCY equipped with a RT scheme allowing to simulate the corresponding TOC reflectances as it is the case for SIF? What datasets would they consider? In situ or space-borne data?

The reviewer is referring to the section: "*The structure of the vegetation affects the observed SIF signal (Magney et al., JGR, 2019; Sun et al., 2023). This is something that could be addressed by including vegetation NIRv, a near-infrared reflectance that has very similar interactions within the canopy to the far-red region SIF. Including NIRv, which can be used to*

*approximate the escape ratio, in the analysis would allow the attribution of the structural effects seen in the SIF signal (Zeng et al., 2019, Dechant et al. 2022).*"

We apologize for the unclear expression here. We aimed to express that by including NIRv in the analysis we could potentially separate the structural influence on the observed signal from the part that is driven by physiological processes. Therefore we would not include it in the model, but instead in the analysis. We have modified the text into

"*...including vegetation NIRv in the analysis... Including NIRv in helping to interpret the SIF signal in the analysis would allow the attribution of the structural effects seen in the SIF signal (Zeng et al., 2019, Dechant et al. 2022). This analysis could be done by using in situ or spaceborn observations.*"

The values of the calculated escape factors could have been compared to literature.

We have removed the calculated escape factors from this version of the manuscript.

77. L564: Could the authors provide references on the variation of SIF emission spectrum across species?

We had a reference to the Magney et al. (JGR, 2019) publication. We have added in new references as suggested by the reviewer (Liu et al. 2025, Magney et al., 2017).

Given that all RT approaches yield approximately the same degree of overestimation in SIF simulations compared to observations, how important is this assumption relative to other potential sources of error? It would be helpful if a ranking of these possible sources of error could be provided.

The assumption that the reviewer is referring to is:
"*This approach has limitations in terms of generalizability, as these spectra differ between species and therefore using a single spectrum for a PFT may introduce uncertainties.*"

One could assume that since the mSCOPE uses a different approach but still ends up with similar overestimation, that this is not causing large error in the estimation of the signal. Other studies have used e.g. spectra generated by SCOPE for this purpose (Beauclaire et al., 2024).

Ranking the different error sources would be very speculative and therefore we're not doing it here.

What is the relative contribution of incorrect parameter values (which could potentially be corrected through data assimilation) compared to incorrect process representation?

This is quite a difficult issue to answer. Based on the leaf level fluorescence yield, we are obtaining similar values as other studies. To be able judge those values, we'd need active observations at site scale (these are on-going at FI-Sod and are planned to be used for this purpose). Are we potentially having errors in our presentation for the radiative transfer of SIF? This is possible, but all the three approaches are giving similar overestimates.

The fact that some other models have given similar results might also suggest that the observations would have e.g. a contribution of soil in the signal. Here a 3D approach would help to disentangle the issue deeper, but those approaches are calculationally too heavy for large scale models.

78. L567: "... in vivo spectrum". Could a reference be provided? I do not understand the implication of this, compared to the other choices made. Could the authors elaborate?

We have added a reference for this to the methods section, where this was introduced.

The reviewer is referring here to:

"*For the L2SM approach we used a theoretical estimate of the in vivo spectrum. This approach would also benefit from further testing with observed spectra.*"

Because we had attenuation inside the leaf included in the L2SM, we couldn't use the observed spectra at the leaf level. In this study we used a theoretical estimate of the in vivo spectrum, as it was more robust than the observed spectrum we had tested earlier. However, this is a topic that would be worthwhile re-visiting in future.

Does this part related to the spectral dependency of SIF really fits within a "radiative transfer" oriented section?

We had considered that it would fit to the section, since the spectral properties influence the radiative transfer. But we also understand other viewpoints on this and have made a new section for this topic.

79. L569-L585: Same remark as before, this part related to data processing do not fit within this RT section (which is already very long).

The text here is related to the potential uncertainties in the observations, and we're e.g. mentioning here that using averaging was useful in this context. Often discussing uncertainties of a study has been part of Discussion, but we'll do our best in replacing part of the context of this section in other parts of the manuscript.

80. L605: Although leaf chlorophyll content is the main factor of leaf absorptance in the visible, this is not the case in the near-infrared.

This is very true and we have now added this point to the statement. It is now:
"*In addition, the amount of leaf chlorophyll influences how much of the SIF emitted from the leaves is attenuated in the canopy in the visible region*".

81. L606-L607: So, this is not the case in QUINCY? What to the authors mean by "include"? Does it relate to the representation of the temporal variations and/or vertical gradients?

The sentence the reviewer is referring to is:

"*Therefore, a model that includes both leaf chlorophyll and SIF included would be beneficial for understanding Earth system processes.*"

We agree that the wording was not optimal, as QUINCY includes leaf chlorophyll. Also it was not clear whether temporal variations or vertical gradients are of interest. We have now modified the sentence to be:

"*Therefore, a model such as QUINCY including both leaf chlorophyll and SIF included would be beneficial for understanding Earth system processes and their temporal variations.*"

**Conclusions:**

82. L615: While the study focuses only on QUINCY TBM, this general statement seem to apply to other similar models. Could the authors elaborate on this? What is expected from a better accounting of the canopy structure by proximal sensing measurements?

We assume that the reviewer is referring to the statement starting in L616, which is:

"*Still, the consistent overestimation of SIF suggests the 3D structure of conifer forest should be better accounted for by models and proximal sensing measurements.*"

The point here is that the two-stream radiative transfer approaches are not optimal to represent radiation fluxes in  coniferous forest. On one side, conifer leaves are needle-shaped, and require specific leaf radiative transfer models (e.g., Liberty (Dawson et al., 1998 )) for which fluorescence emission has not been implemented. Models such as PROSPECT (Jacquemoud and Baret, 1990) and models, where fluorescence emission has been implemented (i.e., FLUSPECT (Vilfan et al., 2016)) represent flat, layered leaves. Moreover, the unidimensional canopy radiative transfer models cannot represent the complex 3D geometry of conifers (cone-shaped). Models like SAIL represent a flat volume of randomly distributed leaves, and therefore, miss the geometry of conifer crowns. Furthermore, conifer leaves clump together, an effect that not all 1D radiative transfer models represent. Uncertainties arise from the varying vertical profile of leaf area index, clumping and self-shading effects.

To further clarify this points, we have modified the text into:

"*We hypothesize that the consistent overestimation might arise from the misrepresentation of conifer leaves and canopy with the commonly used 1D canopy, and leaf plate-theory-based radiative transfer models. However, no fluorescence emission has been implemented in needle-like leaf radiative transfer models, and 3D canopy transfer modules are too computationally demanding for TBMs. Thus, simpler modeling solutions should be explored to improve the representation of fluorescence emission in conifer forests, with the help of proximal sensing measurements.*"

83. L619: I thought that sustained NPQ was not applied at Sodankylä... A clearer presentation of the simulation configurations is needed.

It was tested at Sodankylä (and shown in Fig. S6). It was not applied when showing the results for the different radiative transfer approaches. We apologize for the unclear representation and will try to make the revised manuscript clearer in this respect.

84. L620: Do this "process" refer to sustained NPQ?

The reviewer is referring here to text:
"*Sustained NPQ was relevant in decoupling the simulated SIF from the observed absorbed PAR and the same parameterization improved model performance at the North American sites but appeared less suitable at the Finnish site. The process is likely linked to the air temperature regime of the sites.*"

We changed "*The process…*" to "*This process…*" and hope that this will clarify the text in this respect.

Why "likely"? Isn t air temperature explicitly used already in the formulation of NPQs?

We had tried to convey here the message that the temperature response of sustained NPQ can be different for sites located within climatically different temperature regimes. E.g. a study by Thum et al. (2009) showed that temperature responses of spring recovery in forests differed between the same species between northern and southern Finland. In the Discussion we had referred to a study by Neri et al. (2024) that stated that photosynthetic capacity of plants depends on the temperature regime of the vegetation's environment.

We have modified the latter sentence into:
"*This process is likely linked to the air temperature regime of the sites, that will be different for sites at different locations.*"

85. L620-L622: TROPOSIF is not a satellite, it is a product derived from the observations by the TROPOMI instrument.

We had here referred to the TROPOSIF product from a satellite as:

"*The satellite TROPOSIF product*"

with the idea that this is a product called TROPOSIF based on satellite data. We apologize for such an unclear wording and have now replaced this by:

"*The TROPOSIF product from satellite*"

What does "seems feasible" mean given that this is exactly what is done in this study (Figure 8)?

This statement was connected to the previous sentence, together being:

"*The TROPOSIF product from satellite was able to capture the low spring values observed at CA-Obs and could therefore probably serve as additional data when implementing the*

*parameterization of sustained NPQ in a global model. Use of TROPOSIF observations additionally in model evaluation and development seems feasible.***"**

The point we tried to convey here, is that the springtime TROPOSIF seems to more follow the observed pattern in SIF and not in absorbed PAR, making the observations potentially useful in trying to parameterize the sustained NPQ. We have now clarified this with an addition:

"*Use of TROPOSIF observations additionally in model evaluation and development seems feasible, given that the springtime behaviour seems to follow better site level observations of SIF than absorbed PAR.*"

86. L625: On the link between SIF and latent heat, I am really unsure that this fits in the conclusion of this manuscript given this has never been addressed earlier. This could be expended in the discussion.

Sure, we remove this sentence and will not go into it in the discussion either in order to not extend the manuscript length further.

87. L627: Given the concerns I have expressed, I have some reservations about the concluding remark. Does it concern QUINCY or does it apply to other TBMs as well?

As mentioned above, we've removed the concluding remark.

**Tables:**

88. Table 1: (°) for the coordinates should be added.

The degree sign has been added to the coordinates, as suggested.

LAI and Air temperature vary with time; Are those mean values?

Apologies for not stating that these are annual mean values for air temperature in the earlier version of the manuscript.

LAI does not change a lot in these ecosystems. However, we added a notion that these are summertime values.

The years considered in the study could be indicated.

They are quite clearly in the text.

89. Table 5: Couldn t the presentation of Table 3 be adopted also for this table (R2 and RMSE on the same line)?

Sure, we'll do this change in Table 5.

**Figures**

90. Figure 1: TROPOSIF should be indicated in place of TROPOMI.

Sure, we'll change that.

The wavelength of the SIF product is 740 nm, not 743.

The wavelength of the product we used was 743 nm.

91. Figure 2: The values on y-axes on the left should also be in purple, not only the label, to facilitate interpretation. The same unit is repeated multiple times; this could be simplified.

We have changed the values on the y-axis on left and removed some of the unit labels.

The years are missing on the left figures.

Thanks for noticing these, we'll add those to the figure or will try to clarify this in the caption.

For improved clarity, it would be more consistent to present the panel labels (a, b, c) prior to their corresponding descriptions.

Sure, we'll do it in the proposed order.

Metrics quantifying the model-data discrepancy (RMSE, R2) could be provided.

Sure, we'll add these.

These remarks apply to all similar figures.

Including metrics in figures showing all the different approaches would be not very clear, therefore we did not add those in these cases.

92. Figure 3 - Figure 7: Please use "LZ" or "Liu/Zeng" or "Zeng" consistently throughout the manuscript and figures.

Apologies for the inconsistency. We have now used "LZ" throughout the figures and tables.

93. Figure 5. Couldn t the different components of KNPQ be also presented?

We will consider adding one figure for the supplement that has these different components.

"photosynthetically active" is missing.
Thanks, we'll add this.

Is it phi_Ft that is shown here (eq 10)?

That's right, thanks for noticing this. We'll correct this in the figure.

94. Figure 6: Dots and crosses are not distinguishable.

We apologize for the unclear presentation. We decided to not have separate symbols in this figure, as mentioned above.

95. Figure 7: "Far-red" should appear more clearly in the corresponding figure insets.

It has been mentioned in the caption and it has already earlier said that this section will only deal with far-red SIF. We'll try to include that to the figure, nevertheless.

Thanks, we'll change that.

740 nm should be considered instead of 743 nm.

The wavelength of the product we used was 743 nm.

What is the "uncertainty" of those TROPOSIF data averaged at 0.5°? Is this a kind of averaged retrieval error or the standard deviation of all individual retrievals ? How does it relates to the "standard deviation" cited in L509?

The uncertainty in the figure had been from the uncertainty of the retrievals. We will clarify this or use standard deviation of the daily retrievals instead, as they likely show higher variability.

What does "TROPOSIF values are daily" mean? It is not clear whether the authors have considered the daily corrected variable in the TROPOSIF product or not.

These were daily averages of the instantaneous values. We have now clarified this in the manuscript.

**References**

Beauclaire, Quentin, Simon De Cannière, François Jonard, Natacha Pezzetti, Laura Delhez, Bernard Longdoz, Modeling gross primary production and transpiration from sun-induced chlorophyll fluorescence using a mechanistic light-response approach, Remote Sensing of Environment, Volume 307, 2024, https://doi.org/10.1016/j.rse.2024.114150.

Dawson T.P., Curran P.J. and Plummer S.E. (1998), LIBERTY - Modeling the effects of leaf biochemical concentration on reflectance spectra, Remote Sensing of Environment, 65(1):50-60

Cheng, Rui *et al* Evaluating photosynthetic activity across Arctic-Boreal land cover types using solar-induced fluorescence; 2022 *Environ. Res. Lett.* **17** 115009 **DOI** 10.1088/1748-9326/ac9dae

Fan, L., Kato, T., Miyauchi, T., Buareal, K., Morozumi, T., & Ono, K. (2025). Data assimilation of solar-induced chlorophyll fluorescence improves gross primary production simulation by a process-based VISIT-SIF model in a rice paddy. *Journal of Geophysical Research: Biogeosciences*, 130, e2024JG008280. https://doi.org/10.1029/2024JG008280

Jacquemoud, S., & Baret, F. (1990). PROSPECT: A model of leaf optical properties spectra. Remote Sensing of Environment, 34(2), 75–91. https://doi.org/10.1016/0034-4257(90)90100-Z

Li, R., Lombardozzi, D., Shi, M., Frankenberg, C., Parazoo, N. C., Köhler, P., et al. (2022). Representation of leaf-to-canopy radiative transfer processes improves simulation of far-red solar-induced chlorophyll fluorescence in the Community Land Model version 5. *Journal of Advances in Modeling Earth Systems*, 14, e2021MS002747. https://doi.org/10.1029/2021MS002747

Liu, Weiwei, Matti Mõttus, Zbyněk Malenovský, Shengwei Shi, Luis Alonso, Jon Atherton, Albert Porcar-Castell, An in situ approach for validation of canopy chlorophyll fluorescence radiative transfer models using the full emission spectrum, Remote Sensing of Environment, Volume 316, 2025, https://doi.org/10.1016/j.rse.2024.114490.

Magney, T.S., Frankenberg, C., Fisher, J.B., Sun, Y., North, G.B., Davis, T.S., Kornfeld, A. and Siebke, K. (2017), Connecting active to passive fluorescence with photosynthesis: a method for evaluating remote sensing measurements of Chl fluorescence. New Phytol, 215: 1594-1608. https://doi.org/10.1111/nph.14662

Miinalainen, T., Ojasalo, A., Croft, H., Aurela, M., Peltoniemi, M., Caldararu, S., Zaehle, S., and Thum, T.: Evaluating the carbon and nitrogen cycles of the QUINCY terrestrial biosphere model using space-born optical remotely-sensed data, Biogeosciences, 22, 6937–6962, https://doi.org/10.5194/bg-22-6937-2025, 2025.

Peano, D., Hemming, D., Delire, C., Fan, Y., Lee, H., Materia, S., Nabel, J. E. M. S., Park, T., Wårlind, D., Wiltshire, A., and Zaehle, S.: Plant phenology evaluation of CRESCENDO land surface models using satellite-derived Leaf Area Index – Part 2: Seasonal trough, peak, and amplitude, Biogeosciences, 22, 7117–7135, https://doi.org/10.5194/bg-22-7117-2025, 2025.

Pierrat, Zoe Amie, Troy Magney, Andrew Maguire, Logan Brissette, Russell Doughty, David R. Bowling, Barry Logan, Nicholas Parazoo, Christian Frankenberg, and Jochen Stutz. 2024. " Seasonal Timing of Fluorescence and Photosynthetic Yields at Needle and Canopy Scales in Evergreen Needleleaf Forests." *Ecology* 105(10): e4402. https://doi.org/10.1002/ecy.4402

Thum, T., Aalto, T., Laurila, T., Aurela, M., Hatakka, J., Lindroth, A., and Vesala, T., 2009. Spring initiation and autumn cessation of boreal coniferous forest $CO_2$ exchange assessed by meteorological and biological variables. Tellus61B, 701-717.

Yang, J.C., Troy S. Magney, Loren P. Albert, Andrew D. Richardson, Christian Frankenberg, Jochen Stutz, Katja Grossmann, Sean P. Burns, Bijan Seyednasrollah, Peter D. Blanken, David R. Bowling: Gross primary production (GPP) and red solar induced fluorescence (SIF) respond differently to light and seasonal environmental conditions in a subalpine conifer

forest, Agricultural and Forest Meteorology, Volume 317, 2022,
https://doi.org/10.1016/j.agrformet.2022.108904.

[Figure]

**Fig. R1.** As Fig. 8 in the manuscript, except that the TROPOSIF is not for the 0.5 degrees x 0.5 degrees area around the site, but instead the observations are within the observed pixel.

---

## Author Comment (AC2)

**Replies to Reviewer 2**

The reviewer comments are in magenta, the replies to the comment are in black.

This manuscript presents a comprehensive effort to implement sun-induced chlorophyll fluorescence (SIF) into the terrestrial biosphere model QUINCY and to evaluate simulated SIF against both tower-based and satellite (TROPOMI) observations at multiple evergreen conifer forest sites. The topic is timely and relevant, as SIF has become an important observational constraint for photosynthesis and gross primary productivity (GPP), yet remains insufficiently represented in many terrestrial biosphere models. The integration of mechanistic radiative transfer and non-photochemical quenching (NPQ) processes within QUINCY represents a valuable step forward. The manuscript is generally well structured and clearly written, and the use of multiple independent observational data sets strengthens the evaluation. However, the novelty of the approach relative to previous SIF-enabled modelling studies is not always clearly articulated, and several methodological aspects require further clarification to ensure reproducibility. In addition, the interpretation of the results could be expanded to better highlight the ecological and physiological implications of the findings. Overall, I find the study promising and suitable for publication after moderate revisions addressing the comments below.

We thank the reviewer for the positive comments on this manuscript and especially for pointing in a very constructive manner to issues where improvements are needed. In the revised manuscript we will pay more attention to the novelty of this work in relation to previous work and will clarify the presentation of the methodology as well as bring in ecological and physiological insights based on our results.

Major Comments

1. Novelty and Positioning within Existing Literature

The manuscript would benefit from a clearer statement of what is new compared to existing approaches that simulate SIF within terrestrial biosphere or land surface models. While the introduction provides a good overview of the importance of SIF, it remains unclear how the present implementation in QUINCY advances beyond previous studies (e.g., SCOPE-based or simplified SIF schemes). I recommend adding a short paragraph explicitly outlining the novel aspects of this work and its main added value.

We thank the reviewer for the insightful comment. In the current work we think the aspects related to novelty would be:

-We have used different radiative transfer schemes for SIF within one model, thus providing a comparison of only these schemes instead of comparing different models (e.g. some studies compare the modelling results of their model to SCOPE results, such as Li et al., 2022)

-Many other studies concentrate only on the far-red region SIF, as most of the current satellite products are located in this wavelength region. In this study we evaluated the model performance for both the red and far-red regions, which can be considered a way to prepare

for the FLEX mission. One could argue that photosystems I and II contribute differently to these wavelength regions of the SIF emission and may exhibit distinct dynamics.(e.g. Porcar-Castell et al., 2021). Studying whether the model performs worse in the other wavelength region in terms of the dynamics could reveal incapability of the model to include relevant processes. Moreover, pronounced differences in biases of the model estimates in the other wavelength region could reveal difficulties in modelling of the radiative transfer of SIF.

-Even though the L2SM model has been published earlier (Quaife, 2025) and used in model study earlier (Knorr et al., 2025, using another leaf level model for SIF), no detailed model evaluation of its performance has been yet shown in the scientific literature.

-Our study includes sites in two different continents. The temperature sensitivity of GPP has been found to be different in the boreal forests of these two different continents (Muccio et al., 2025) and in our study we can assess whether the temperature dependent processes, such as sustained non-photochemical quenching are different at sites located in different continents. The study by Chen et al. (2024) also included a similar set-up concerning sites, but there the third site was located in South Korea.

-Two of the tested radiative transfer approaches include attenuation of the SIF signal in the leaf. A comparison of the magnitudes of these approaches with an approach that omits this process will aid in evaluating its importance.

We will write a short paragraph to the introduction that makes clearer the novelty aspects of the current work.

**2. Description of the SIF Implementation**

The description of the SIF module in the Methods section is relatively high level. For a modelling study, additional technical detail is needed. For example, the authors could consider providing more justification for the chosen parameter values and indicating whether they are site-specific, plant functional type-specific, or globally fixed. You can also add typical parameter value ranges in Table 2. You can also consider adding a schematic diagram summarizing the SIF calculation within the QUINCY framework. These additions would significantly improve transparency and reproducibility.

We thank the reviewer for this very good improvement suggestion. We will write more clearly how we chose the parameter values and how they were applied in this study (they were all global in our case, as we were only working with one plant functional type, but unfortunately this was not made clear in the first version of the manuscript).

Many of the items in Table 2 are actually variables (apologies for the unclear caption, which we have now clarified). Adding a schematic diagram is a good idea and also suggested by the other reviewer. We will add conceptual figures clarifying the calculation of the SIF.

**3. Model Evaluation and Metrics**

The comparison with tower-based and TROPOMI SIF is a strong aspect of the study. However, the evaluation would benefit from a more consistent quantitative assessment.

Please report standard performance metrics (e.g., RMSE, bias, correlation coefficient) consistently across all sites and modelling configurations, for example, Figures 2-5. In addition, the scale mismatch between tower measurements and satellite pixels should be discussed more explicitly, particularly in the interpretation of model–TROPOMI comparisons.

We will add a more consistent quantitative assessment of the TROPOMI SIF comparison. We will add standard performance metrics in the figure 2. The metrics will be added in a common table for all the model configurations at the three sites. We will also add a table that will show the metrics and the improvement caused by the implementation of the sustained NPQ with the L2SM radiative transfer approach at all sites (shown in Figure 5 for CA-Obs site).

We will also add discussion on the scale mismatch between the tower measurements and satellite pixels. We will additionally add a figure for the TROPOMI-model comparison at FI-Sod.

**4. Interpretation of Results**

The Results section focuses mainly on model performance, but the Discussion could be strengthened by deeper interpretation of the findings. For example, what do the results imply about the seasonal regulation of photosynthesis in evergreen conifer forests? How does the inclusion of NPQ affect the relationship between SIF and GPP across seasons? Are the differences among sites indicative of meaningful ecological or climatic controls? Expanding on these points would broaden the relevance of the study.

We thank the reviewer for these insightful remarks that are worth addressing in the discussion.

The fact that the same "state of acclimation" parameterization in slowing the development of spring recovery in GPP across the three sites is successful shows that the GPP would be well enough constrained with the same parameterization at these sites.

Using the same parameterization for sustained non-photochemical quenching did not show to be at first that successful at FI-Sod. However, the overestimation of simulated absorbed PAR might be partly contributing to this. We added in the figure showing the APAR estimation from the FloX box also an APAR estimation based on above- and below canopy PAR sensors. These observations started on June 18th 2021, so unfortunately these will not help to untangle the model mismatch taking place in early June. However, they will help to illustrate how the impact of the footprint mismatch contributes to the model performance (similarly to what was seen in the diurnal cycle plot in Fig. S8d of simulated and observed SIF at FI-Sod). Adding the plot for the TROPOSIF vs. simulations will help to assess the sustained NPQ formulation at the FI-Sod site better.

We will do some additional analysis, such as looking into the temperature response of chlorophyll fluorescence yield at the sites with and without the sustained NPQ formulation (similarly to what has been done by Kim et al., 2021) and also look into the SIF/GPP ratio across seasons as well as a function of temperature (as done by Chen J. et al., 2022 and Chen, R. et al., 2025). These analyses will help us to address the interesting points that the reviewer was raising here and we will discuss their results in depth.

5. Uncertainties and Limitations

Please discuss the main sources of uncertainty in both the model and the observations (e.g., uncertainty in tower-based SIF retrievals, satellite noise, parameter uncertainty). A clearer discussion of model limitations would help place the results in context.

In the first version of the manuscript we had some discussion on the uncertainty of the observations, but since it did not have its own subsection, it was not that clear. We have now made a subsection for the observational errors and expanded that to account for the noise in the satellite observations.

We will also add a section for the model uncertainties and also add a clearer discussion of model limitations.

Minor Comments:

1. Table 1 could be improved by adding time period information of SIF measurements.

Thanks, we have now added this information to Table 1.

2. Ensure that all acronyms are defined at first use. For example, TBMs were defined multiple times in L35, L39 and L145.

We apologize for sloppiness with this issue. We will pay attention to correct this in the revised manuscript version.

3. Improve figure readability (font sizes, legend clarity) and consider expanding figure captions. For example, the font sizes of Figure 4 and 6 are too small.

Thanks for this remark. We'll work on all the figures to make them more readable and expanding the figure captions.

4. Minor language edits are recommended to improve clarity and conciseness in a few sections.

Thanks, we'll go through the whole manuscript and aim for clarity and conciseness in the language.

**References**

Chen, R., Liu, L., Liu, Z., Liu, X., Kim, J., Kim, H. S., Lee, H., Wu, G., Guo, C., & Gu, L. (2024). SIF-based GPP modeling for evergreen forests considering the seasonal variation in maximum photochemical efficiency. Agricultural and Forest Meteorology, 344,. https://doi.org/10.1016/j.agrformet.2023.109814

Chen, R., L. Liu, X. Liu, C. Y. S. Wong and I. Ensminger, "Temperature-Dependent Relationship Between Solar-Induced Chlorophyll Fluorescence and Photosynthesis in

Evergreen Needleleaf Forests," in IEEE Transactions on Geoscience and Remote Sensing, vol. 63, pp. 1-11, 2025, Art no. 4423011, doi: 10.1109/TGRS.2025.3620306.

Chen, J.; Liu, X.; Ma, Y.; Liu, L. Effects of Low Temperature on the Relationship between Solar-Induced Chlorophyll Fluorescence and Gross Primary Productivity across Different Plant Function Types. Remote Sens. 2022, *14*, 3716. https://doi.org/10.3390/rs14153716

Kim, J., Ryu, Y., Dechant, B., Lee, H., Kim, H. S., Kornfeld, A., & Berry, J. A. (2021). Solar-induced chlorophyll fluorescence is non-linearly related to canopy photosynthesis in a temperate evergreen needleleaf forest during the fall transition*. Remote Sensing of Environment, 258, 112362. https://doi.org/10.1016/j.rse.2021.112362

Muccio, D., Keppel-Aleks, G., & Parazoo, N. (2025). Contrasting temperature sensitivity of boreal forest productivity in North America and Eurasia. *Journal of Geophysical Research: Biogeosciences*, 130, e2024JG008634. https://doi.org/10.1029/2024JG008634